# The p97/VCP segregase is essential for arsenic-induced degradation of PML and PML-RARA

Ellis G. Jaffray[1], Michael H. Tatham[1], Barbara Mojsa[1], Magda Liczmanska[1], Alejandro Rojas-Fernandez[1], Yili Yin[1], Graeme Ball[1], and Ronald T. Hay[1]

**Acute Promyelocytic Leukemia is caused by expression of the oncogenic Promyelocytic Leukemia (PML)–Retinoic Acid Receptor Alpha (RARA) fusion protein. Therapy with arsenic trioxide results in degradation of PML-RARA and PML and cures the disease. Modification of PML and PML-RARA with SUMO and ubiquitin precedes ubiquitin-mediated proteolysis. To identify additional components of this pathway, we performed proteomics on PML bodies. This revealed that association of p97/VCP segregase with PML bodies is increased after arsenic treatment. Pharmacological inhibition of p97 altered the number, morphology, and size of PML bodies, accumulated SUMO and ubiquitin modified PML and blocked arsenic-induced degradation of PML-RARA and PML. p97 localized to PML bodies in response to arsenic, and siRNA-mediated depletion showed that p97 cofactors UFD1 and NPLOC4 were critical for PML degradation. Thus, the UFD1-NPLOC4-p97 segregase complex is required to extract poly-ubiquitinated, poly-SUMOylated PML from PML bodies, prior to degradation by the proteasome.**

## Introduction

The Promyelocytic Leukemia protein (PML) was first recognized in Acute Promyelocytic Leukemia (APL) where it is translationally fused to the retinoic acid receptor α (RARA) as a consequence of the t (15;17) chromosomal translocation (de The et al., 1990; Kakizuka et al., 1991). Linkage of PML to RARA generates a PML-RARA oncoprotein that acts as a transcription factor that deregulates transcriptional programs blocking the differentiation of hematopoietic progenitor cells and thus causing leukemia (Grignani et al., 1993; Kwok et al., 2006; Martens et al., 2010; Mikesch et al., 2010; Tan et al., 2021). Alternative splicing of the PML mRNA generates seven isoforms (I–VII) that differ in their C-terminal regions. The common N-terminal region contains a conserved Tripartite Motif (TRIM) that is composed of a RING domain, two zinc coordinating B-boxes, and a coiled coil that facilitates dimerization (Bernardi and Pandolfi, 2007). It is this region of PML fused to RARA which is the hallmark of APL. PML is also known as TRIM19 and belongs to a large family of TRIM proteins. In normal cells, PML is dynamically associated with discrete nuclear structures named PML nuclear bodies (PML NBs), which can also recruit additional proteins including SUMO, p53, Daxx, and Sp100 (Brand et al., 2010). However, in cells expressing the PML-RARA oncoprotein, it is dispersed into a larger number of smaller nuclear bodies (Bernardi and Pandolfi, 2007; Dyck et al., 1994; Weis et al., 1994). APL was once a disease with a very poor prognosis, but most patients are now cured by treatment with a combination of arsenic trioxide and all trans retinoic acid (Lo-Coco et al., 2013; Mi et al., 2015; Wang and Chen, 2008). This treatment targets the oncoprotein for degradation, thus relieving the transcriptional repression, allowing the accumulated promyelocytes to terminally differentiate (Dos Santos et al., 2013; Salomoni, 2009). It is known that arsenic triggers rapid, multisite SUMO modification of PML and PML-RARA (Muller et al., 1998). Ubiquitination of SUMO modified PML is catalyzed by Ring finger protein 4 (RNF4), a member of the family of SUMO Targeted Ubiquitin Ligases (STUbLs; Geoffroy and Hay, 2009; Perry et al., 2008). Initially isolated as a co-activator of the androgen receptor (Moilanen et al., 1998), RNF4 colocalizes with SUMO and PML in nuclear bodies (Hakli et al., 2005). Human RNF4 is a protein of 194 amino acids that contains multiple SUMO interaction motifs (SIMs; Tatham et al., 2008) and a RING domain (Hakli et al., 2004). Under normal conditions RNF4 is monomeric and inactive, but the high local concentration of SUMO in PML bodies generated by arsenic treatment mediates SIM-dependent recruitment of RNF4 and drives RING domain dimerization and activation of RNF4 (Rojas-Fernandez et al.,

[1]Centre for Gene Regulation and Expression, School of Life Sciences, University of Dundee, Dundee, UK.

Correspondence to Ronald T. Hay: r.t.hay@dundee.ac.uk

M. Liczmanska's current affiliation is Ubiquigent Ltd. Dundee Technopole, James Lindsay Place, Dundee, UK.    A. Rojas-Fernandez's current affiliation is Instituto de Medicina & Centro Interdisciplinario de Estudios del Sistema Nervioso, Universidad Austral de Chile, Valdivia, Chile.

2014). Therefore, PML that has undergone extensive arsenic-induced SUMO modification and RNF4 mediated ubiquitination is targeted for ubiquitin-mediated proteolysis via the proteasome (Lallemand-Breitenbach et al., 2008; Lallemand-Breitenbach et al., 2001; Tatham et al., 2008). Cells in which RNF4 expression has been abrogated accumulate PML protein and arsenic administration does not lead to degradation of PML. As a consequence, SUMO-modified PML accumulates in nuclear bodies (Lallemand-Breitenbach et al., 2008; Tatham et al., 2008).

To allow identification of additional components of this pathway we carried out proteomic analysis of PML bodies isolated from untreated and arsenic-treated cells. This revealed that the p97/valosin-containing protein (VCP) segregase (Meyer and Weihl, 2014; Ye et al., 2017) displayed an increased association with PML bodies after arsenic treatment. Pharmacological inhibition of p97 ATPase activity blocked arsenic-induced degradation of PML while immunofluorescence analysis indicated that p97 was recruited into PML bodies in response to arsenic. siRNA-mediated depletion of nuclear cofactors of p97 indicated that UFD1 and NPLOC4 were also required for the arsenic-induced degradation of PML. Following p97 inhibition, SUMO and ubiquitin modified PML accumulated, indicating that they are intermediates in a degradation pathway that involves p97. In patient-derived NB4 cells that express the PML-RARA oncogene, pharmacological inhibition of p97 blocked arsenic-induced degradation of both PML and PML-RARA. Both proteins were degraded through the proteasome rather than via autophagy. These studies have uncovered an additional step in the pathway of arsenic-induced degradation of PML and PML-RARA, in which the p97 segregase is required to extract the ubiquitinated proteins from PML bodies prior to ubiquitin mediated proteolysis.

## Results

### Proteomic analysis identifies p97 as a PML body-associated protein

As a tool to identify additional factors involved in the arsenic-induced degradation of PML, CRISPR/cas9 was used to generate U2OS PML$^{-/-}$ cells, which were then reconstituted with YFP tagged PMLV (Fig. S1 A)—herein referred to as YFP-PML. YFP-PML is expressed at similar levels to endogenous PML in WT U2OS cells, is localized in typical PML NBs, and undergoes post-translational modification in response to arsenic (Fig. S1, A and B). To identify factors associated with PML bodies that alter in abundance during arsenic exposure a proteomic experiment using these cells was undertaken (Fig. 1 A). U2OS PML$^{-/-}$ + YFP-PML and U2OS WT cells (as a control) were either untreated or exposed to 1 µM arsenic for 2 h in quadruplicate cultures (Fig. S1 B). Cells were lysed under conditions that preserved SUMO and ubiquitin modifications, isolated nuclei were disrupted by sonication, followed by benzonase treatment and nuclear debris removed by centrifugation. This released intact PML bodies as judged by microscopy using YFP fluorescence (Fig. S1, C and D). PML bodies were isolated from the supernatant using a GFP nanobody coupled to magnetic beads. This procedure efficiently captured YFP-PML but did not capture untagged PML from

U2OS cells (Fig. S1 E). PML was detected in eluates from beads incubated in extracts from U2OS PML$^{-/-}$ + YFP-PML cells, but not in WT U2OS cells (Fig. S1 F). YFP-PML from untreated cells was largely unmodified and arsenic treatment resulted in the formation of a higher molecular weight species of YFP-PML (Fig. S1, E and F). For proteomic analysis the elutions were fractionated by SDS-PAGE (Fig. S2 A) and digested with trypsin before identification and label-free quantification by mass spectrometry (MS). For the 2,156 protein groups identified, principal component analysis of the log-transformed intensity data indicated clear separation between WT U2OS and U2OS$^{-/-}$PML + YFP-PML purifications and between treated and untreated samples (Fig. S2 B). Putative PML interactors were shortlisted by comparing relative abundance between anti-GFP nanobody purifications from U2OS PML$^{-/-}$ + YFP-PML and WT U2OS cells (Fig. S2 C). To further filter our list of putative PML body components for proteins likely to be relevant to arsenic-induced PML degradation, we also compared relative protein abundance between anti-GFP nanobody purifications from arsenic-treated and untreated U2OS PML$^{-/-}$ + YFP-PML cells (Fig. S2 D). In total, 196 proteins were enriched in nanobody purifications from U2OS PML$^{-/-}$ + YFP-PML cells (false discovery rate [FDR] 0.05%), while 87 proteins increased in response to arsenic (FDR 10%). A control comparison using purifications from U2OS WT cells (Fig. S2 E) only found one protein to significantly differ by the same statistical parameters. 21 proteins made the cut-off for both comparisons, i.e., being significantly enriched in YFP-PML purifications and increased in abundance after arsenic treatment (Fig. S2 F). These proteins are illustrated on a scatter plot where the abundance ratio between anti-GFP nanobody purifications from U2OS PML$^{-/-}$ + YFP-PML and WT U2OS cells are plotted against the abundance ratio between anti-GFP nanobody purifications from U2OS PML$^{-/-}$ + YFP-PML that were either untreated or treated with arsenic (Fig. 1 B).

Cell-biological studies and immunofluorescence analyses have identified numerous proteins that colocalize with PML bodies (documented in the Human Protein Atlas [Thul et al., 2017]) and more recently proximity labeling has identified potential PML interactors. The overlap of these two sets with our list is indicated in Fig. 1 C and Data S1. Also, several proteins associated with nucleoli were included in our list of PML interacting proteins (Data S1). We do not believe that this is a result of nucleolar contamination or post-lysis association of nucleoli. Rather, this likely reflects a dynamic association of some nucleolar proteins with PML bodies. This is exemplified by the SUMO targeted AAA ATPase MDN1 and the SUMO substrate PELP1 that are reported to reside in nucleoli (Raman et al., 2016), but are included in our list of PML associated proteins (Fig. 1 C). By immunofluorescence we confirmed the dual localization of PEPL1 and MDN1, while in contrast, the abundant protein fibrillarin was exclusively nucleolar (Fig. S2 G). As SUMO and ubiquitin modifications in PML bodies increases in response to arsenic, it was not surprising that these proteins and enzymes for their conjugation were present in this small group (Fig. 1 C). The p97/VCP segregase was an unexpected member of this group of outliers (Fig. 1, B–D). The presence of p97 and other potential PML interactors was confirmed by Western blotting of

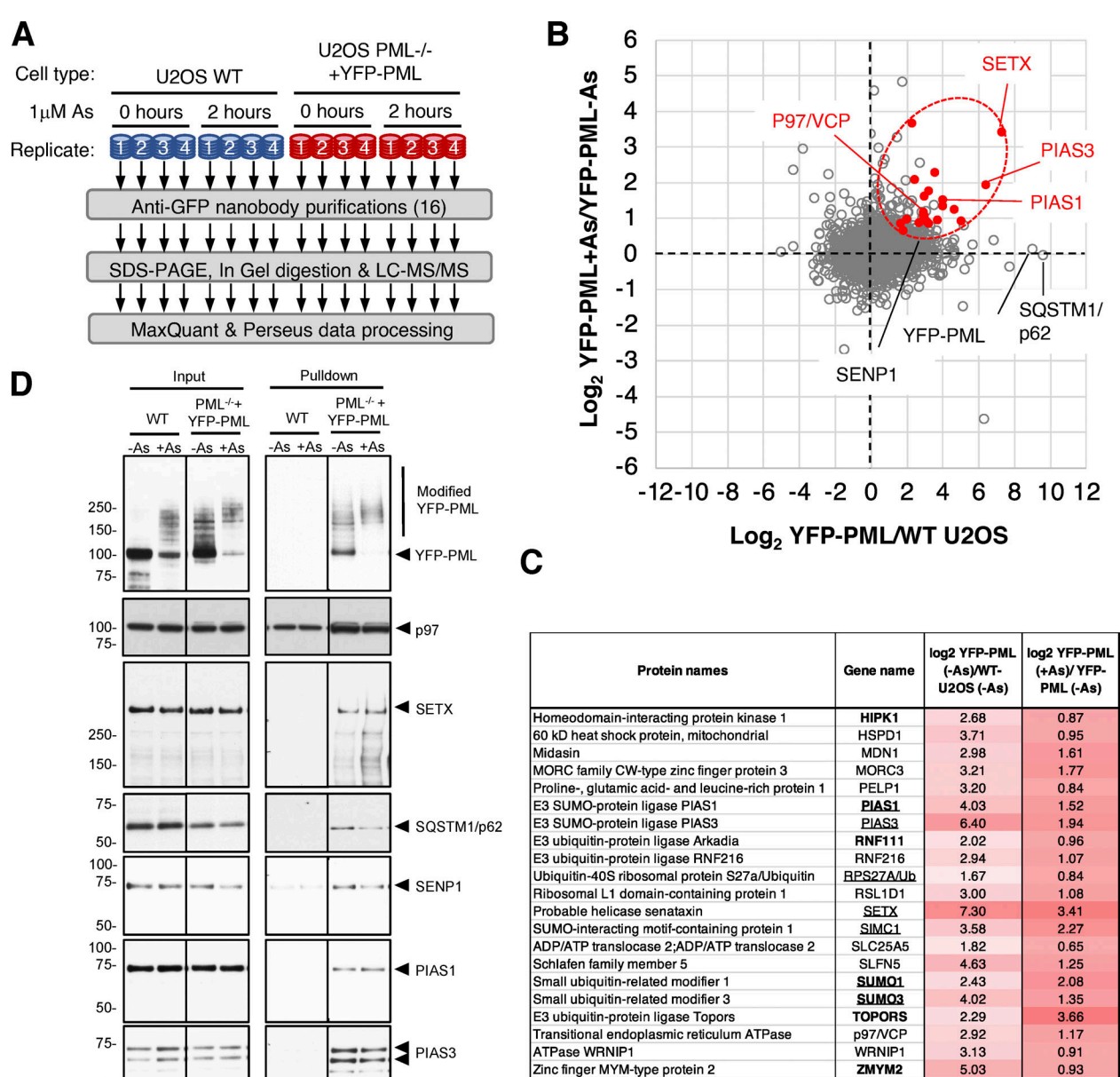

C

| Protein names | Gene name | log2 YFP-PML (-As)/WT-U2OS (-As) | log2 YFP-PML (+As)/ YFP-PML (-As) |
|---|---|---|---|
| Homeodomain-interacting protein kinase 1 | **HIPK1** | 2.68 | 0.87 |
| 60 kD heat shock protein, mitochondrial | HSPD1 | 3.71 | 0.95 |
| Midasin | MDN1 | 2.98 | 1.61 |
| MORC family CW-type zinc finger protein 3 | MORC3 | 3.21 | 1.77 |
| Proline-, glutamic acid- and leucine-rich protein 1 | PELP1 | 3.20 | 0.84 |
| E3 SUMO-protein ligase PIAS1 | **PIAS1** | 4.03 | 1.52 |
| E3 SUMO-protein ligase PIAS3 | PIAS3 | 6.40 | 1.94 |
| E3 ubiquitin-protein ligase Arkadia | **RNF111** | 2.02 | 0.96 |
| E3 ubiquitin-protein ligase RNF216 | RNF216 | 2.94 | 1.07 |
| Ubiquitin-40S ribosomal protein S27a/Ubiquitin | RPS27A/Ub | 1.67 | 0.84 |
| Ribosomal L1 domain-containing protein 1 | RSL1D1 | 3.00 | 1.08 |
| Probable helicase senataxin | SETX | 7.30 | 3.41 |
| SUMO-interacting motif-containing protein 1 | SIMC1 | 3.58 | 2.27 |
| ADP/ATP translocase 2;ADP/ATP translocase 2 | SLC25A5 | 1.82 | 0.65 |
| Schlafen family member 5 | SLFN5 | 4.63 | 1.25 |
| Small ubiquitin-related modifier 1 | **SUMO1** | 2.43 | 2.08 |
| Small ubiquitin-related modifier 3 | **SUMO3** | 4.02 | 1.35 |
| E3 ubiquitin-protein ligase Topors | **TOPORS** | 2.29 | 3.66 |
| Transitional endoplasmic reticulum ATPase | p97/VCP | 2.92 | 1.17 |
| ATPase WRNIP1 | WRNIP1 | 3.13 | 0.91 |
| Zinc finger MYM-type protein 2 | **ZMYM2** | 5.03 | 0.93 |

Figure 1. **Proteomic analysis of arsenic-induced PML body associated proteins. (A)** Overview of the proteomic experimental design. For each condition, $n = 4$ separate experiments. **(B)** Scatter plot of $log_2$ ratios comparing protein abundance in anti-GFP nanobody purifications from YFP-PML expressing $PML^{-/-}$ U2OS cells with those from WT U2OS cells (x-axis) and comparing abundance in purifications form YFP-PML cells during 1 μM arsenic treatment with the same purifications from cells not exposed to arsenic (y-axis). Data for 2,157 proteins shown. Red markers are proteins defined as significantly different in both comparisons (see Fig. S2). **(C)** Summary data for the 21 proteins marked red in B. Underlined gene names were identified previously as PML proximal proteins (Barroso-Gomila et al., 2021), and bold entries are described in the Human Protein Atlas (Thul et al., 2017; http://proteinatlas.org) as belonging to PML body. **(D)** Anti-GFP nanobody purifications were prepared from the indicated cells after 6 h 1 μM arsenic (+As) or untreated (−As). Both inputs and bead elutions were Western blotted for the indicated species. Molecular weight markers are indicated in kD. Source data are available for this figure: SourceData F1.

PML associated material (Fig. 1 D). Current models of PML degradation suggest that arsenic-induced SUMO modification of PML leads to RNF4 mediated ubiquitination followed by proteasomal degradation. A role for p97 in this pathway has not been documented.

### Pharmacological inhibition of p97 interferes with arsenic-induced degradation of PML

To determine whether p97 plays a role in arsenic-induced degradation of PML, we made use of CB-5083, a potent and highly selective inhibitor of the ATPase activity of the D2 domain of p97 (Anderson et al., 2015; Zhou et al., 2015). U2OS $PML^{-/-}$ + YFP-PML cells were treated with arsenic, CB-5083, or a combination of both over a 4-h period and analyzed by Western blotting (Fig. 2 A). In untreated cells, a broad distribution of PML species is evident over a range of 100–250 kD. Treatment with arsenic for 1 h shifts more PML into the highest molecular weight region of the gel, presumably because of SUMO and ubiquitin modifications (Tatham et al., 2008). After 4 h, PML levels begin to reduce, again presumably due to degradation (Fig. 2 A).

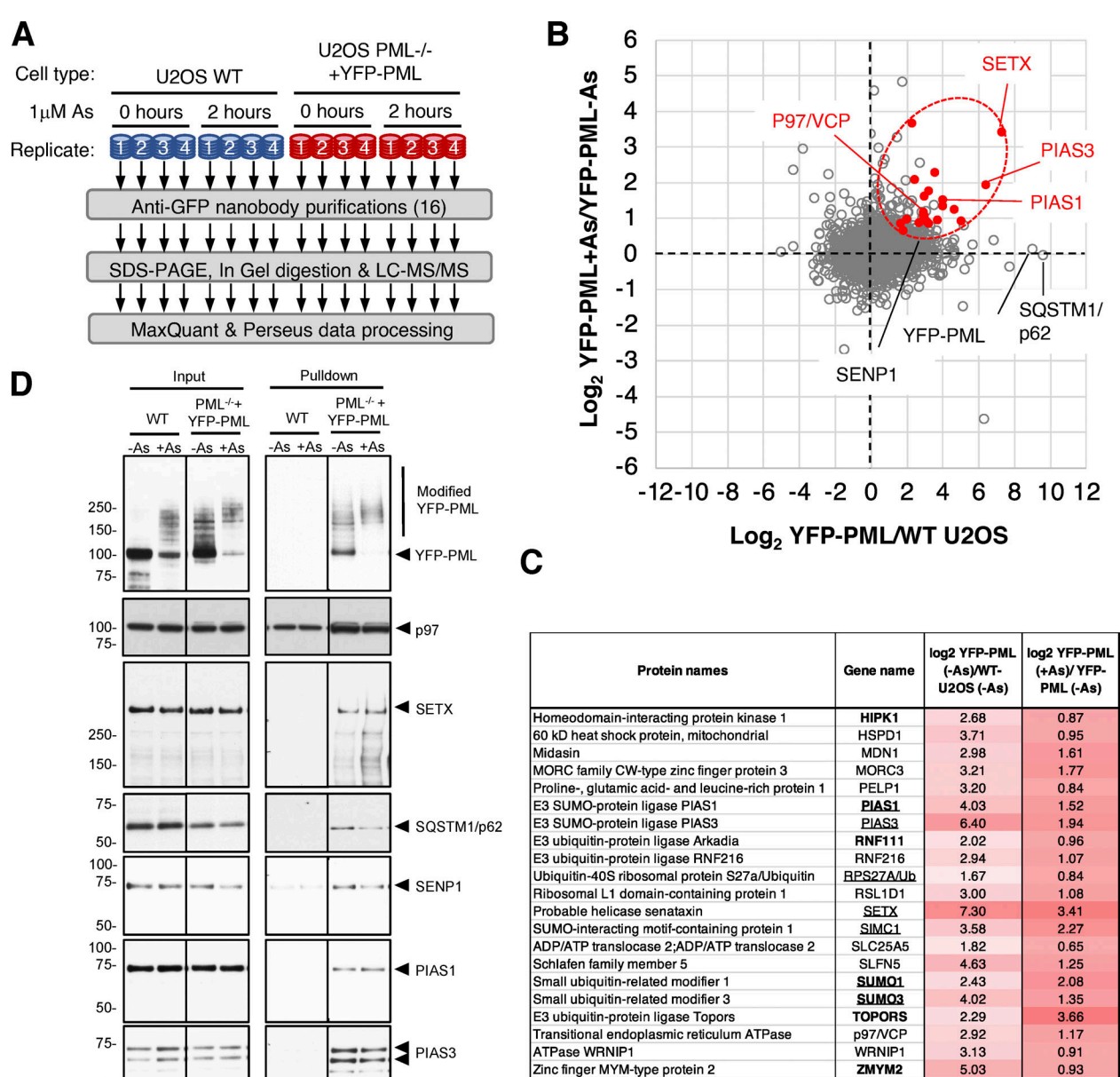

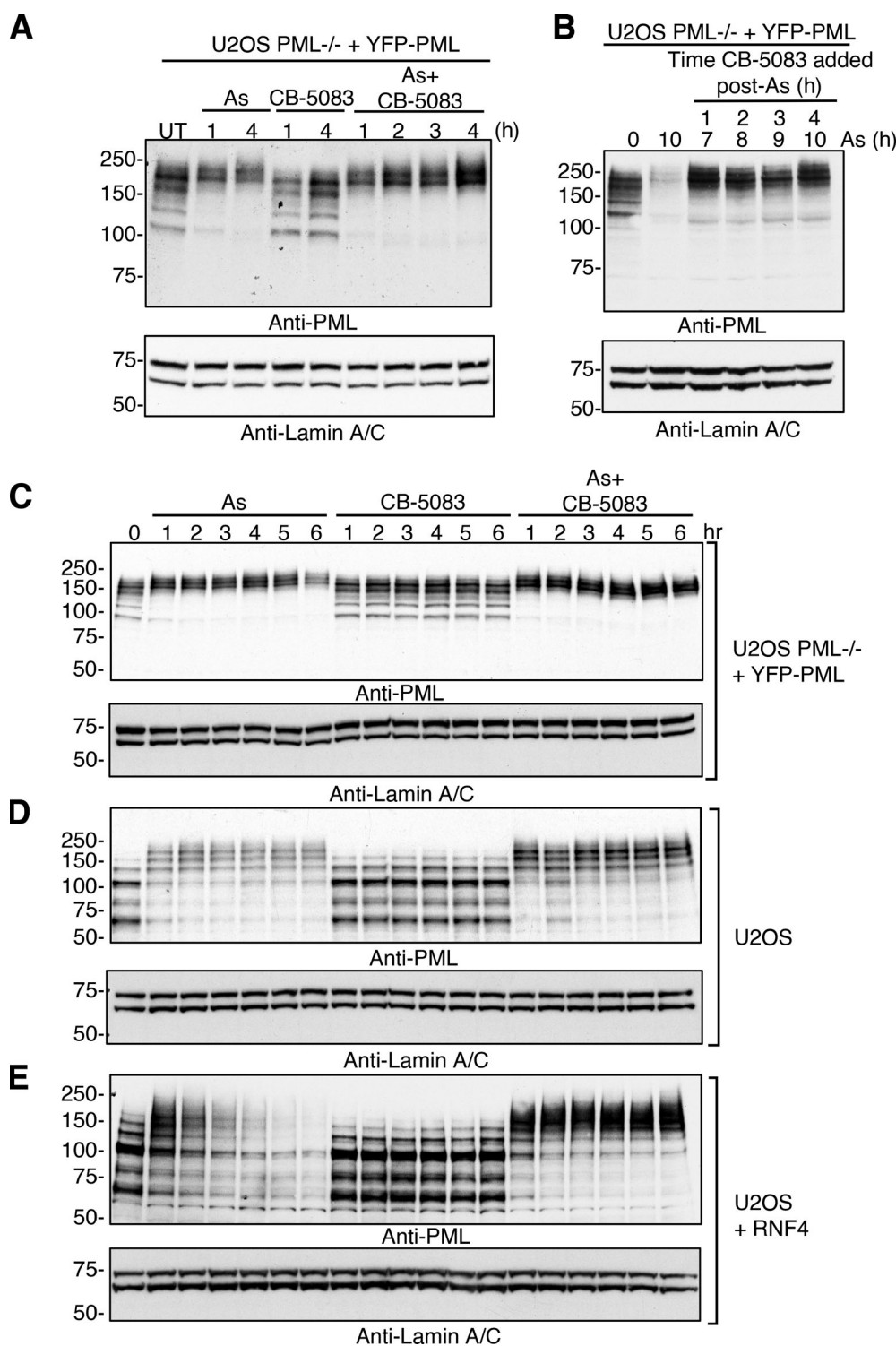

Figure 2. **Pharmacological inhibition of p97 interferes with arsenic-induced degradation of PML. (A)** U2OS PML$^{-/-}$ + YFP-PML cells were either untreated (UT) or treated with 1 µM arsenic (As), 5 µM CB-5083, or a combination of arsenic and CB-5083 for the times indicated. Cell lysates were analyzed by Western blotting using an antibody to PML or Lamin A/C. **(B)** U2OS PML$^{-/-}$ + YFP-PML cells were treated with 1 µM arsenic for the indicated times and 5 µM CB-5083 added at indicated times after the start of arsenic treatment. Cell lysates were analyzed by Western blotting using an antibody to PML or Lamin A/C. **(C–E)** U2OS PML$^{-/-}$ + YFP-PML cells (C), U2OS cells (D), or U2OS + RNF4 cells (E) were either untreated (UT) or treated with 1 µM arsenic (As), 5 µM CB-5083, or a combination of arsenic and CB-5083 for the times indicated. Cell lysates were analyzed by Western blotting using an antibody to PML or Lamin A/C. Molecular weight markers are indicated in kD. Source data are available for this figure: SourceData F2.

Treatment of the cells with CB-5083 alone does not alter the molecular weight distribution of PML, although treatment with arsenic and CB-5083 results in the accumulation of PML in higher molecular weight species and inhibition of the PML degradation (Fig. 2 A). To further resolve the kinetics of the CB-5083 effect, arsenic was first added to five cultures of U2OS PML$^{-/-}$ + YFP-PML cells and then CB-5083 was added 1, 2, 3, or 4 h later and incubated for a further 6 h, or no CB-5083 was added and the cells incubated for 10 h. Cells treated with arsenic alone for 10 h displayed almost complete degradation of PML, whereas this degradation was blocked, even when CB-5083 was added 4 h after arsenic (Fig. 2 B).

To investigate the effect of p97 inhibition on the arsenic-induced degradation of multiple isoforms of endogenous PML, WT U2OS cells overexpressing RNF4 were used. RNF4 overexpression was required to accelerate PML ubiquitination, thus avoiding long-term exposure of cells to CB-5083, which is known to be toxic (Le Moigne et al., 2017). As described above, arsenic induces SUMO modification and substantial degradation of YFP-PML after 6 h and this is blocked by CB-5083, resulting in the accumulation of higher molecular weight species of PML (Fig. 2 C). Likewise, all isoforms of endogenous PML in U2OS cells shift to higher molecular weight forms and show evidence of arsenic-induced degradation, which is blocked by CB-5083 (Fig. 2 D). RNF4 overexpression in U2OS cells accelerates arsenic-induced degradation of PML to the point where most PML species have been degraded after 6 h. This degradation was completely blocked by CB-5083 with all PML accumulating as higher molecular weight species (Fig. 2 E). In all cases, treatment with CB-5083 alone did not alter the electrophoretic mobility of PML species (Fig. 2, C–E).

### Inhibition of p97 alters the arsenic-induced changes to PML body number, morphology, and content

Arsenic-induced degradation of PML is associated with dramatic changes to the number, morphology, and content of PML bodies (Geoffroy et al., 2010; Jeanne et al., 2010). To determine how these parameters were affected by p97 inhibition, the influence of CB-5083 upon arsenic-induced PML degradation was investigated by time-lapse microscopy in U2OS PML$^{-/-}$ + YFP-PML cells. 10 videos were collected for each condition and an automated quantitative pipeline was established to follow PML body size, number of PML bodies per cell, total PML body area, and total YFP-PML intensity over time (Videos 1, 2, 3, and 4). Stills from the videos at 0 and 6.75 h (Fig. 3 A) show there was little change in any parameters measured in untreated cells, although arsenic treatment reduced the number and overall intensity of PML bodies, which was blocked by CB-5083. It was also apparent that CB-5083 alone lead to an increase in PML body size (Fig. 3 A). This was confirmed by quantitation of PML body size, which showed no change in untreated or arsenic-treated cells, but increased in cells treated with CB-5083, irrespective of whether they were also treated with arsenic (Fig. 3 B). The number of PML bodies per cell is reduced in response to arsenic, but this reduction is less apparent in the presence of CB-5083 (Fig. 3 B). The two metrics PML body area per cell and PML intensity per cell both represent the amount of PML present in

nuclear bodies, and showed PML levels reduced in response to arsenic, which is completely blocked in the presence of CB-5083 (Fig. 3 B). Statistical analysis of PML body intensity at the 6.75 h time-point shows a significant reduction in PML with arsenic treatment, which is abrogated by CB-5083 treatment (Fig. 3 C).

### P97 is recruited to PML bodies in response to arsenic

Pharmacological inhibition with CB-5083 suggests that p97 plays a role in arsenic-induced degradation of PML and proteomic analysis suggests p97 co-purifies with PML in PML-enriched nuclear fractions. To confirm p97 is indeed recruited to PML bodies, we examined U2OS PML$^{-/-}$ + YFP-PML cells by fluorescence microscopy for up to 6 h after arsenic treatment. p97 recruitment was evaluated by determining the extent of colocalization between PML marked by YFP fluorescence and staining with an antibody specific to p97 (Fig. S2 H). As p97 is a highly abundant protein involved in many different cellular processes, we employed a pre-extraction procedure to release the bulk of the soluble p97 from the cells before fixation. Although PML bodies reduce in size and number after exposure to arsenic (Fig. 3), it was necessary to select cells still containing PML to allow study of p97 colocalization over the whole time course. This revealed that in untreated cells p97 did not strongly colocalize with PML bodies, whereas arsenic treatment brought about a time-dependent increase in the association of p97 with PML bodies (Fig. 4 A). The percentage of PML bodies colocalized with p97 was determined at 0, 2, 4, and 6 h after arsenic treatment (Fig. 4 B). In untreated cells, about 15% of PML bodies were associated with p97, which increased to almost 60% after 2 h arsenic treatment, 75% after 4 h, and 95% after 6 h (Fig. 4 B). siRNA-mediated depletion of RNF4, TAK243-mediated inhibition of ubiquitination, and ML792-mediated inhibition of SUMOylation confirmed the requirement for SUMO-mediated, RNF4-dependent ubiquitination for p97 recruitment to PML bodies (Fig. S2 H). In contrast, siRNA depletion (Table 1) of the p97 associated metallo-protease SPRTN, which is involved in clearing protein crosslinks from DNA (Gibbs-Seymour et al., 2015), did not appear to affect p97 recruitment (Fig. S2 H). To confirm the association of p97 with PML bodies, we made use of a p97 variant with an E578Q mutation in the Walker B motif of the D2 domain that traps substrate proteins (Hulsmann et al., 2018). U2OS WT or U2OS PML$^{-/-}$ + YFP-PML cells were transfected with either Myc-tagged p97 WT or E578Q and were treated with arsenic for 6 h or were untreated. We used the anti-GFP nanobody affinity purification method described above to enrich cellular extracts for YFP-PML and analyzed them for the presence of PML and Myc-tagged p97 by Western blotting. YFP-PML was efficiently isolated, but untagged PML from U2OS cells was not (Fig. 4 C). Under these conditions, the E578Q substrate-trapping mutant of p97 displayed enhanced association with YFP-PML compared to WT p97 (Fig. 4 C), particularly after arsenic treatment, indicating that PML is likely to be a direct substrate of p97.

### p97 cofactors UFD1 and NPLOC4 are required for arsenic-induced degradation of PML

p97 is typically recruited to ubiquitinated proteins by cofactors containing ubiquitin binding domains. Among the large number

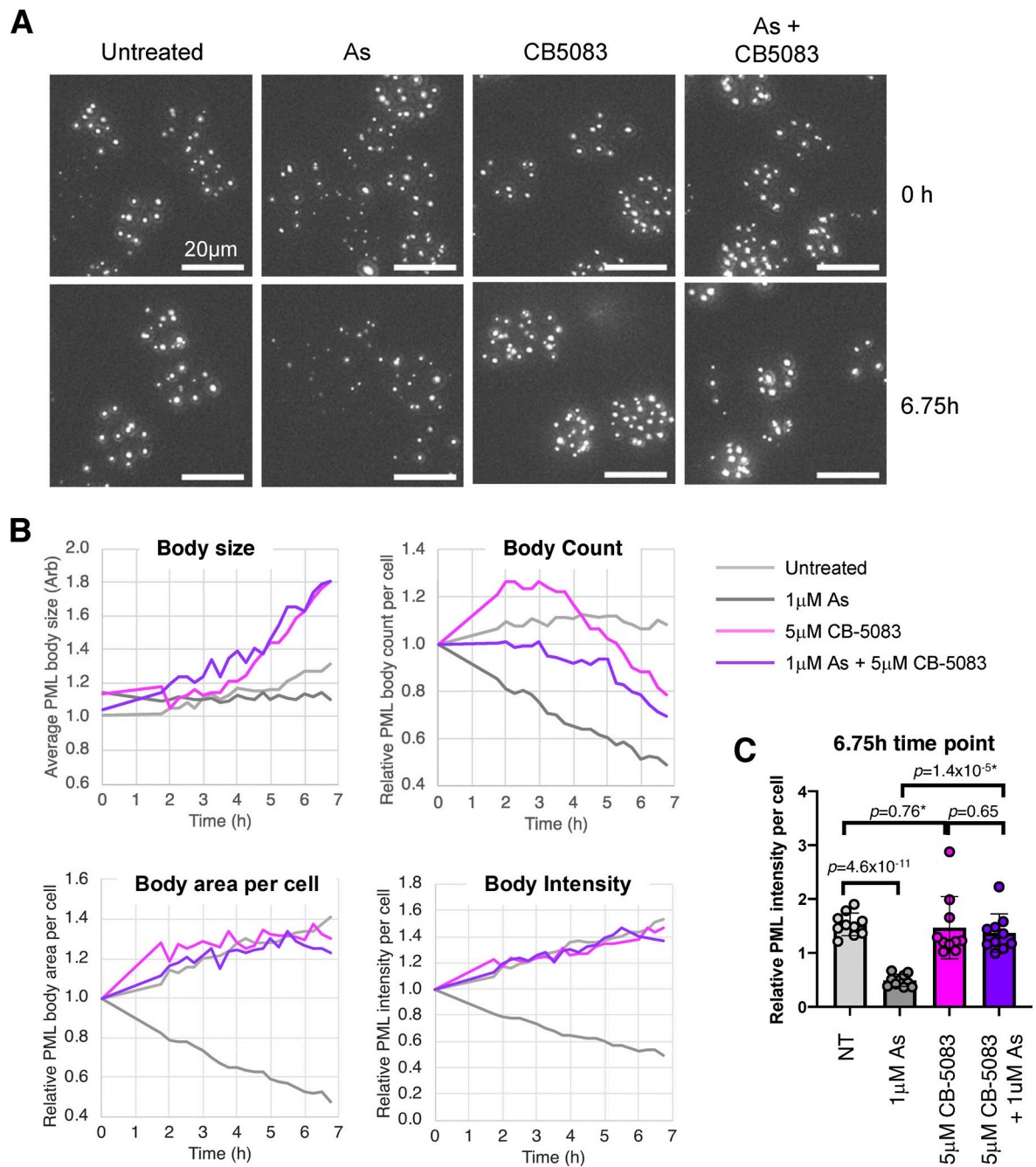

Figure 3. **Inhibition of p97 alters arsenic-induced changes to PML body number, morphology, and content.** U2OS PML$^{-/-}$ + YFP-PML cells were either untreated or treated with 1 μM arsenic (As), 5 μM CB-5083, or a combination of arsenic and CB-5083 and analyzed by time-lapse fluorescent imaging for 6.75 h. 10 videos were collected for each condition (one video from each condition is shown in Videos 1, 2, 3, and 4). **(A)** Stills from the start (0 h) and the end (6.75 h) of the movies. Scale bars are 20 μm. **(B)** The average PML body size, number of PML bodies per cell, total area of PML bodies, and total intensity of PML bodies was plotted as a function of time (n = 10 fields of view). **(C)** Comparison of relative PML intensity per cell after 6.75 h treatment as indicated. Columns represent average and error bars are 1 SD. P values are two-tailed student t tests assuming equal variance or unequal variance (*). Each data point is derived from a single field of view taken from the time course shown in B (n = 10). Source data are available for this figure: SourceData F3.

of p97 cofactors involved in many different cellular processes, we focused on those that were reported to function in the nucleus. siRNAs to UFD1, NPLOC4, UBXN6, FAF2, and PLAA were used to deplete the respective proteins in U2OS PML$^{-/-}$ + YFP-PML cells. In this background, arsenic-induced degradation of PML over a 24-h period was followed by Western blotting (Fig. S3 A). In all cases, knockdown of the target protein was effective,

although as expected, knockdown of NPLOC4 also led to the depletion of its heterodimeric partner UFD1 (Fig. S3 B). In cells treated with non-targeting (NT) siRNA, arsenic exposure led to the accumulation of higher molecular weight PML forms which were reduced after 6 h and almost completely depleted after 24 h. While knockdown of UBXN6, FAF2, and PLAA failed to block PML degradation, knockdown of UFD1 and NPLOC4 led to

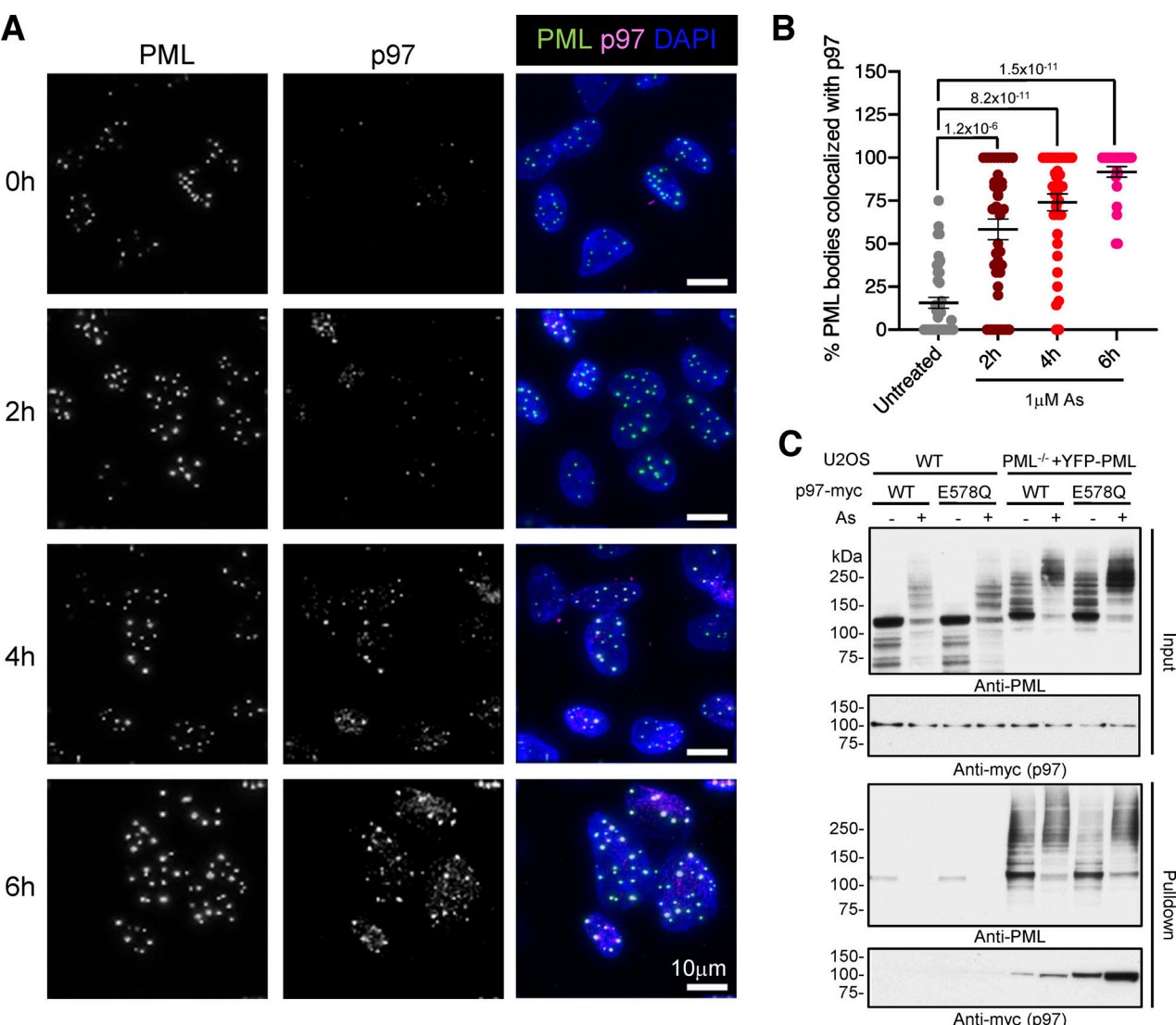

Figure 4. **P97 is recruited to PML bodies in response to arsenic.** U2OS PML$^{-/-}$ + YFP-PML cells were treated with arsenic for the times indicated, pre-extracted as described in Materials and methods, fixed, and stained with an antibody to p97. **(A)** Images of p97 (green), YFP-PML (red), and DAPI (blue) are shown. Scale bars are 10 µm. **(B)** The percentage of PML bodies colocalized with p97 was plotted against time of arsenic treatment. Chart shows individual cell data. Bars represent mean with SEM. P values are two-tailed Mann-Whitney U test derived by comparison with untreated. Cell count n = 41 (untreated), n = 38 (2 h), n = 38 (4 h), and n = 25 (6 h). **(C)** WT U2OS cells or PML$^{-/-}$ + YFP-PML U2OS cells were transfected with plasmids expressing myc-p97 constructs of either WT or E578Q mutant form and either treated or not with 1 µM arsenic for 6 h. Nuclear extracts (Input) or YFP-PML purifications (Pulldown) were blotted for PML or Myc. Molecular weight markers are indicated in kD. Source data are available for this figure: SourceData F4.

reduced degradation of PML and accumulation of higher molecular weight forms of PML, even after 24 h of arsenic treatment (Fig. S3 A). To confirm these findings and determine the consequences of siRNA-mediated depletion of p97 and associated cofactors on PML body number, size, and PML content, U2OS PML$^{-/-}$ + YFP-PML cells were treated with siRNA for 48 h before exposure to arsenic for a further 24 h. Cells were analyzed by Western blotting at 0, 6, and 24 h after arsenic addition and by fluorescence microscopy after 0 and 24 h. Even prior to arsenic treatment, Western blotting shows that knockdown of p97 leads to the accumulation of PML (Fig. 5 A). Although less pronounced, this is also evident with knockdown of UFD1 and NPLOC4. Unexpectedly, knockdown of UBXN6 leads to a reduction in PML levels prior to arsenic treatment, suggesting it

has a positive influence on PML body numbers under normal conditions (Fig. 5 A and Fig. S3 A). In response to arsenic, knockdown of p97, UFD1, and NPLOC4 appears to block PML degradation. In contrast, knockdown of UBXN6 appears to enhance arsenic-induced degradation of PML (Fig. 5 A). Analysis of YFP-PML by fluorescence microscopy indicates that cells treated with NT siRNA show the expected reduction in PML bodies in response to 24 h arsenic treatment. In contrast, cells treated with siRNA to p97, UFD1, and NPLOC4 and exposed to arsenic accumulate PML in large intense nuclear bodies, which is not the case for UBXN6 (Fig. 5 B). To quantify arsenic-induced PML degradation after siRNA treatment, time-lapse microscopy was used to monitor YFP-PML fluorescence over a 10-h period. 10 videos were collected for each siRNA treatment and the

Table 1. **SiRNA details**

| siRNA name | Target sequence |
| --- | --- |
| J-006557-07, RNF4 | 5'-GCUAAUACUUGCCCAACUUUU-3' |
| J-006557-08, RNF4 | 5'-GAAUGGACGUCUCAUCGUUUU-3' |
| J-006557-09, RNF4 | 5'-GACAGAGACGUAUAUGUGAUU-3' |
| J-006557-10, RNF4 | 5'-GCAAUAAAUUCUAGACAAGUU-3' |
| J-008727-09, VCP | 5'-GCAUGUGGGUGCUGACUUA-3' |
| J-008727-10, VCP | 5'-CAAAUUGGCUGGUGAGUCU-3' |
| J-008727-11, VCP | 5'-CCUGAUUGCUCGAGCUGUA-3' |
| J-008727-12, VCP | 5'-GUAAUCUCUUCGAGGUAUA-3' |
| J-008785-09, UBXN6 | 5'-UCAUGAAGAUCUACACGUU-3' |
| J-008785-10, UBXN6 | 5'-AGUUCUACGUGCUGAGCGA-3' |
| J-008785-11, UBXN6 | 5'-GUGUUUCAGGAGCGCAUUA-3' |
| J-008785-12, UBXN6 | 5'-AAAGGAACUUCAAGCCGAA-3' |
| J-010649-09, FAF2 | 5'-GGACCUAACUGACGAAUGA-3' |
| J-010649-18, FAF2 | 5'-UGUACAGGACAGAUUGAAU-3' |
| J-010649-19, FAF2 | 5'-AGGAUAACCUUACGUGAAA-3' |
| J-010549-20, FAF2 | 5'-CCUAAUGAUUCUCGAGUAG-3' |
| J-016215-05, PLAA | 5'-GCGAGUGUCUUGAAGUAUA-3' |
| J-016215-06, PLAA | 5'-GUAAGUGAAUGCUGUAGAU-3' |
| J-016215-07, PLAA | 5'-GGAAGUAGUACAAGACCUA-3' |
| J-016215-08, PLAA | 5'-UGUAAGAGGUUUGGCAAUU-3' |
| J-017918-05, UFD1L | 5'-GAGCGUCAACCUUCAAGUG-3' |
| J-017918-06, UFD1L | 5'-GGUAAGAUAACUUUCAUCA-3' |
| J-017918-07, UFD1L | 5'-UGACAUGAACGUGGACUUU-3' |
| J-017918-08, UFD1L | 5'-CCCAAAGCCGUAUUAGAAA-3' |
| J-020796-09, NPLOC4 | 5'-AGGAAAAGCAUUGGCGAUU-3' |
| J-020796-10, NPLOC4 | 5'-AUAAUGGCUUCUCGGUUUA-3' |
| J-020796-11, NPLOC4 | 5'-ACUCUGAAGUUCCGGCAUU-3' |
| J-020796-12, NPLOC4 | 5'-CUGAAGUGGCUGCGAUUUA-3' |

quantitative pipeline described above was used to follow PML bodies over time (Videos. 5, 6, 7, 8, 9, and 10). In cells treated with NT siRNA, YFP-PML was depleted in a time-dependent fashion in response to arsenic (Fig. 5 C), reaching >80% depletion after 9.75 h (Fig. 5 D). siRNA to the cofactor UBXN6 had no apparent effect. In cells treated with siRNA to p97, UFD1, or NPLOC4, the rate of YFP-PML depletion was reduced (Fig. 5 C), with final depletion after 9.75 h reaching only 40% for siRNA to p97 and 60% for siRNAs for UFD1 or NPLOC4. In all measured PML body metrics, UBXN6 depletion had no significant effect, showing similar data to non-treated cells for PML body numbers, size, and area after arsenic treatment (Fig. S4, A–C). However, cells exposed to siRNA for UFD1 or NPLOC4 showed blunted responses to arsenic treatment by the same measurements (Fig. S4, A–C). This was also true for siRNA for p97, except in this case PML body area increased upon arsenic treatment. These data suggest that the heterodimer of UFD1 and NPLOC4 is the cofactor that is required for arsenic-induced degradation of PML.

## P97 inhibition leads to the accumulation of arsenic-induced degradation intermediates

To identify the arsenic-induced post-translational modifications that accumulate on PML when p97 is inhibited, we used the anti-GFP nanobody affinity purification method described above to enrich cellular extracts for YFP-PML and analyzed them for the presence of PML, SUMO1, SUMO2/3, and ubiquitin by Western blot. To evaluate the modification status of PML under different conditions, the bead-bound material was treated with recombinant proteases to specifically release ubiquitin or SUMO molecules from PML. Beads were left untreated, treated with the catalytic domain of the SUMO-specific protease SENP1, treated with the catalytic domain of the ubiquitin specific protease USP2, or treated with both SENP1 and USP2. After treatment, the beads were washed, and the eluted material was subjected to Western blotting. In samples untreated by proteases, PML levels were reduced in response to arsenic and all the PML migrated as a higher molecular weight smear up the gel (Fig. 6 A). The distribution and abundance of PML species after CB-5083 treatment were not significantly different from that of untreated cells. However, the combination of arsenic and CB-5083 treatment stabilized the YFP-PML and lead to the appearance of a more defined high molecular weight species (Fig. 6 A). Treatment of the beads with SENP1 returned most of these heavier forms back to a more rapidly migrating species consistent with unmodified YFP-PML, although some slowly migrating forms of PML persisted (Fig. 6 A). Treatment with USP2 had little impact on the distribution of PML species, indicating that most of the more slowly migrating material was a consequence of SUMO modification. Blotting with an antibody against SUMO1 indicates that in untreated cells there are relatively low levels of SUMO1 associated with PML, while after arsenic treatment this increases dramatically (Fig. 6 B). CB-5083 treatment alone has no impact on SUMO1 modification, but the combination of arsenic and CB-5083 leads to an accumulation of SUMO1 (Fig. 6 B), presumably because of the block to arsenic-induced degradation of PML. As expected, these species disappear upon SENP1 treatment, but are not altered by treatment with USP2 (Fig. 6 B). The arsenic-induced change to SUMO2 modification is less dramatic than SUMO1, but still increases, and again this accumulates further in the additional presence of CB-5083 (Fig. 6 C). SUMO2 is also completely removed by SENP1, but not by USP2 (Fig. 6 C). Blotting with an antibody to ubiquitin reveals that it is present at low levels on PML from untreated cells but is dramatically increased after arsenic treatment and further accumulated in the additional presence of CB-5083 (Fig. 6 D). Treatment with SENP1 removes a substantial proportion of the ubiquitin, with the remaining ubiquitin reactivity now associated with species of a reduced molecular weight (Fig. 6 D), representing ubiquitin directly conjugated to PML. Most of the ubiquitin is removed by treatment with USP2, although in this case the enzyme digestion has not gone to completion (Fig. 6 D). A requirement for p97 extraction appears to be a substrate modified by at least a pentamer of K48-linked ubiquitin (Deegan et al., 2020; Twomey et al., 2019). To search for such an intermediate, we blotted with an antibody specific for K48-linked ubiquitin. K48 chains associated with PML were detected, but

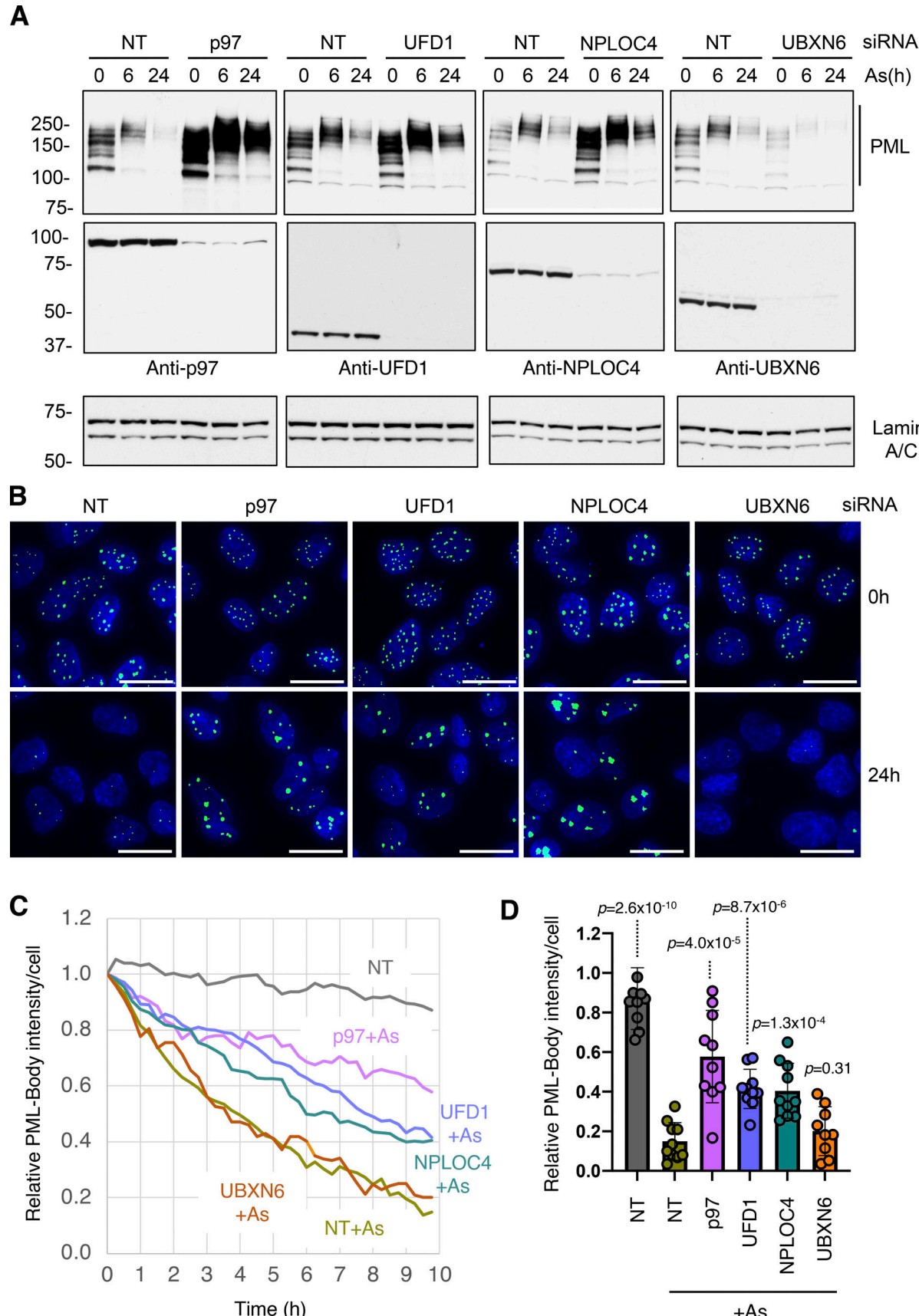

Figure 5. **p97 cofactors UFD1 and NPLOC4 are required for arsenic-induced degradation of PML.** U2OS PML$^{-/-}$ + YFP-PML cells were treated with NT siRNA or siRNAs to p97 cofactors and exposed to arsenic. **(A)** Western blot analysis with antibodies to PML, p97 cofactors, or Lamin A/C. Molecular weight

markers are indicated in kD. **(B)** Immunofluorescence analysis of YFP-PML in response to arsenic. Scale bars are 20 µm. **(C)** Time-lapse fluorescent imaging for 9.75 h after exposure to arsenic. 10 videos were collected for each condition (n = 10; example videos in Videos 5, 6, 7, 8, 9, and 10). Average relative PML body intensity per cell was plotted versus time. **(D)** Relative PML intensity per cell of cells treated with NT or siRNAs to p97 cofactors after 10 h arsenic (As) treatment. P values are comparisons with NT + As using two-tailed unpaired Student's t test assuming similar deviations. Each data point is a single field of view taken from C (n = 10). Source data are available for this figure: SourceData F5.

only in the presence of the p97 inhibitor CB-5083 (Fig. 6 E). Unlike the bulk of ubiquitin, treatment with SENP1 does not appear to collapse the K48-linked PML to lower molecular weight species, suggesting that the K48 ubiquitin chains are directly linked to PML. We also detected association of K63 chains with PML, but this was not stabilized by the p97 inhibitor (Fig. 6 F). Although undetected in the Western blot experiment, we did observe an arsenic-induced increase in K48 chains in the absence of p97 inhibition by MS (Fig. 6 G). K63 chains associated with PML were detected by MS, but the arsenic-induced increase was not statistically significant (Fig. 6 G). These data indicate that arsenic induces complete modification of PML by SUMO, with a greater response by SUMO1 than SUMO2. Also, the subsequent ubiquitination takes place mostly on the SUMO molecules, with a smaller fraction of K48-linked chains directly conjugated to PML. All these intermediates in the pathway accumulate in the presence CB-5083, presumably by blocking extraction from PML bodies by p97, which would normally occur prior to proteasomal degradation.

### Arsenic-induced degradation of the PML-RARA oncoprotein requires p97

The therapeutic value of arsenic in the treatment of APL is its ability to induce degradation, not only of PML but also the PML-RARA oncoprotein. It is therefore important to establish whether PML-RARA is degraded through the same p97-dependent pathway as PML. We utilized the patient-derived NB4 cells (Lanotte et al., 1991) that harbor the t (15;17) chromosomal translocation responsible for expression of the PML-RARA oncogenic fusion protein. These cells also express WT forms of PML and RARA. NB4 cells were treated with arsenic for 0, 4, 6, 8 h and either untreated or treated with 4 µM CB-5083. Whole-cell extracts were analyzed by Western blotting using antibodies against PML or RARA. Blotting of untreated cells with antibodies to PML reveals species with a wide range of molecular weights that presumably include PML-RARA, WT PML isoforms derived from differential splicing, and SUMO/ubiquitin modified species (Fig. 7 A). After 4 h of treatment with arsenic, all these species migrate with a decreased electrophoretic mobility and after 8 h have reduced in intensity as a consequence of arsenic-induced degradation. In the presence of 4 µM CB-5083, arsenic still induces a shift in PML to higher molecular weight, but the subsequent degradation is blocked (Fig. 7 A). Blotting with an antibody to RARA reveals that the WT form of RARA does not change in response to arsenic or CB-5083 (Fig. 7 A), with the PML-RARA fusion protein being present as multiple more slowly migrating forms. In response to arsenic, PML-RARA shifts to higher molecular weight forms that undergo degradation. In the additional presence of CB-5083, the shift to higher molecular weight species is also apparent, but these persist as

the arsenic-induced degradation is blocked (Fig. 7 A). Levels of p97 did not change in response to any of these treatments (Fig. 7 A).

Analysis of NB4 cells at 0, 2, and 6 h after treatment with arsenic alone, CB-5083 alone, or arsenic plus CB-5083 by immunofluorescence using antibodies to PML and RARA revealed that in untreated cells both were in very small atypical nuclear bodies (Fig. S5). However, after 2 h arsenic treatment, both PML and RARA accumulated in larger bodies before starting to undergo degradation after 4 h (Fig. S5). Treatment with CB-5083 was without consequence over the 4-h period, whereas in the presence of arsenic plus CB-5083, both PML and RARA accumulated in larger nuclear bodies, but did not undergo degradation (Fig. S5).

These experiments reveal that PML and PML-RARA both require the action of the p97 segregase for degradation in response to arsenic. However, it is not clear whether the p97 segregase functions as a prelude to degradation via the proteasome or via autophagy. To distinguish between these possibilities, we made use of bafilomycin, an inhibitor of autophagy, and bortezomib, an inhibitor of the proteasome. NB4 cells were untreated or treated with arsenic either alone or in the additional presence of CB-5083, bafilomycin, or bortezomib. Western blotting of whole-cell extracts with antibodies to PML and RARA revealed that PML and PML-RARA were both degraded in the presence of arsenic, but in the additional presence of CB-5083 or bortezomib, high molecular weight species of both PML and PML-RARA accumulated, and degradation was blocked (Fig. 7 B). In contrast, the addition of bafilomycin did not alter the course of arsenic-induced degradation of either PML or PML-RARA (Fig. 7 B). Treatment with CB-5083, bafilomycin, or bortezomib in the absence of arsenic did not alter the patterns of PML or PML-RARA species detected by Western blotting. To confirm the activity of bafilomycin, the autophagic marker LC3 was found to accumulate in the presence of bafilomycin irrespective of the presence of arsenic (Fig. 7 B). Blotting with an antibody to ubiquitin revealed that it accumulated in higher molecular weight forms and free ubiquitin was depleted in the presence of bortezomib (Fig. 7 B).

## Discussion

Treatment with arsenic has emerged as a highly successful therapy for APL (Lallemand-Breitenbach et al., 2012) as it induces degradation of both PML and the PML-RARA oncogene that cause the disease. As arsenic targets the PML component of PML-RARA, it is thought that the pathway of arsenic-induced degradation was identical for PML and PML-RARA and that it involved SUMO-dependent, ubiquitin-mediated proteasomal degradation (Lallemand-Breitenbach et al., 2008; Tatham et al.,

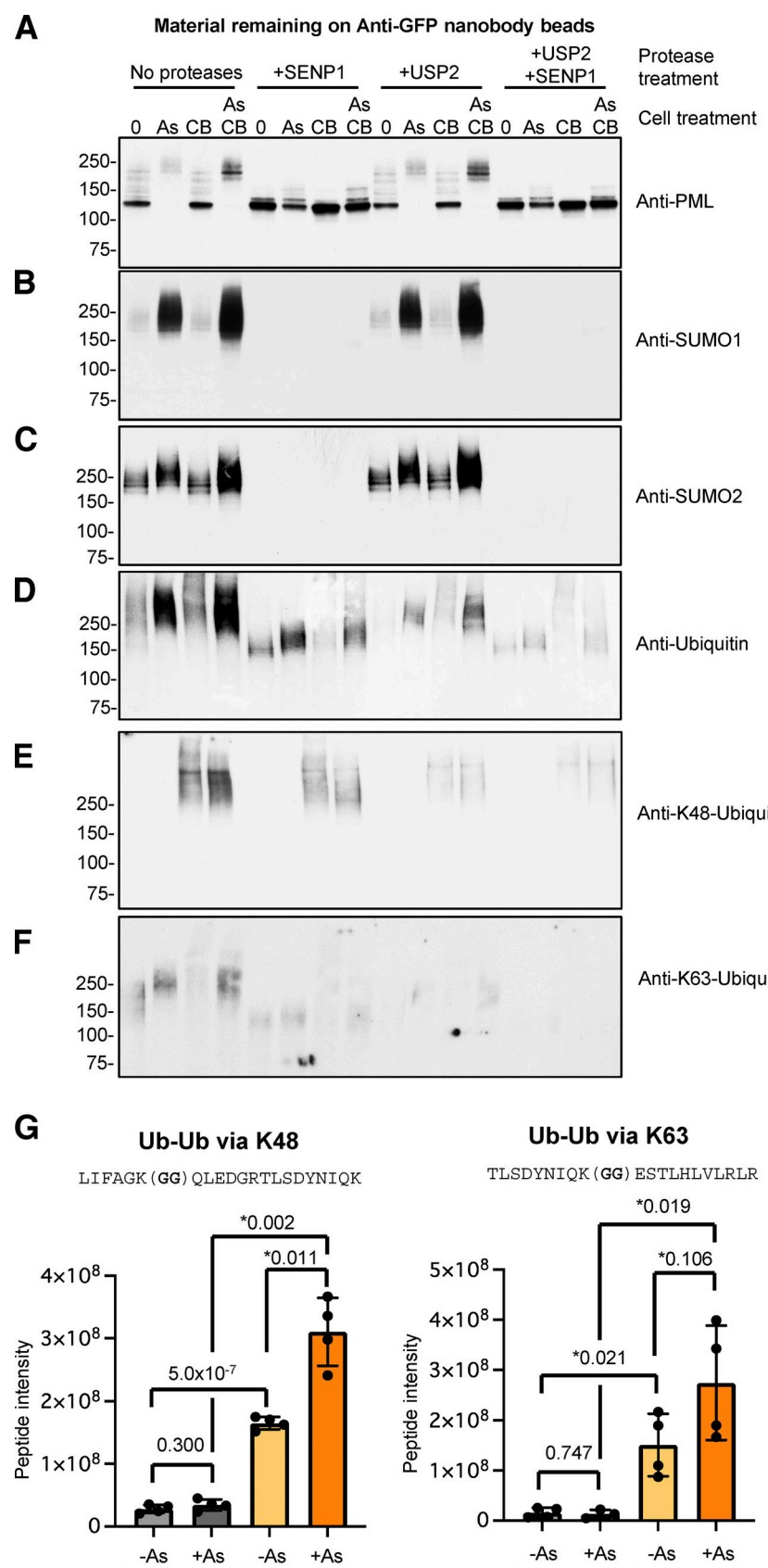

Figure 6. **P97 inhibition leads to the accumulation of arsenic-induced degradation intermediates.** PML bodies were isolated from U2OS PML[−/−] + YFP-PML cells that were either untreated (0), treated with 1 µM arsenic (As), 5 µM CB-5083 (CB), or a combination of arsenic and CB-5083 (As/CB). Purified PML bodies bound to anti-GFP nanobody magnetic beads were either untreated (no proteases), treated with SUMO specific protease SENP1 (+SENP1), treated with ubiquitin-specific protease USP2 (+USP2), or a combination of both (+USP2+SENP1). **(A–F)** Bound material was eluted and analyzed by Western blotting using antibodies to PML (A), SUMO1 (B), SUMO2 (C), ubiquitin (D), K48-linked ubiquitin (E), K63-linked ubiquitin (F). Molecular weight markers are indicated in kD. **(G)** Relative intensity in the indicated proteomics samples of peptides characteristic of Ub-Ub branch points at K48 (left) and K63 (right). Bars show average intensity and errors are 1 SD. P values from Students two-tailed *t* test (*with Welch's correction) are shown above bars. *n* = 4 independent experiments (see Fig. 1 A). Source data are available for this figure: SourceData F6.

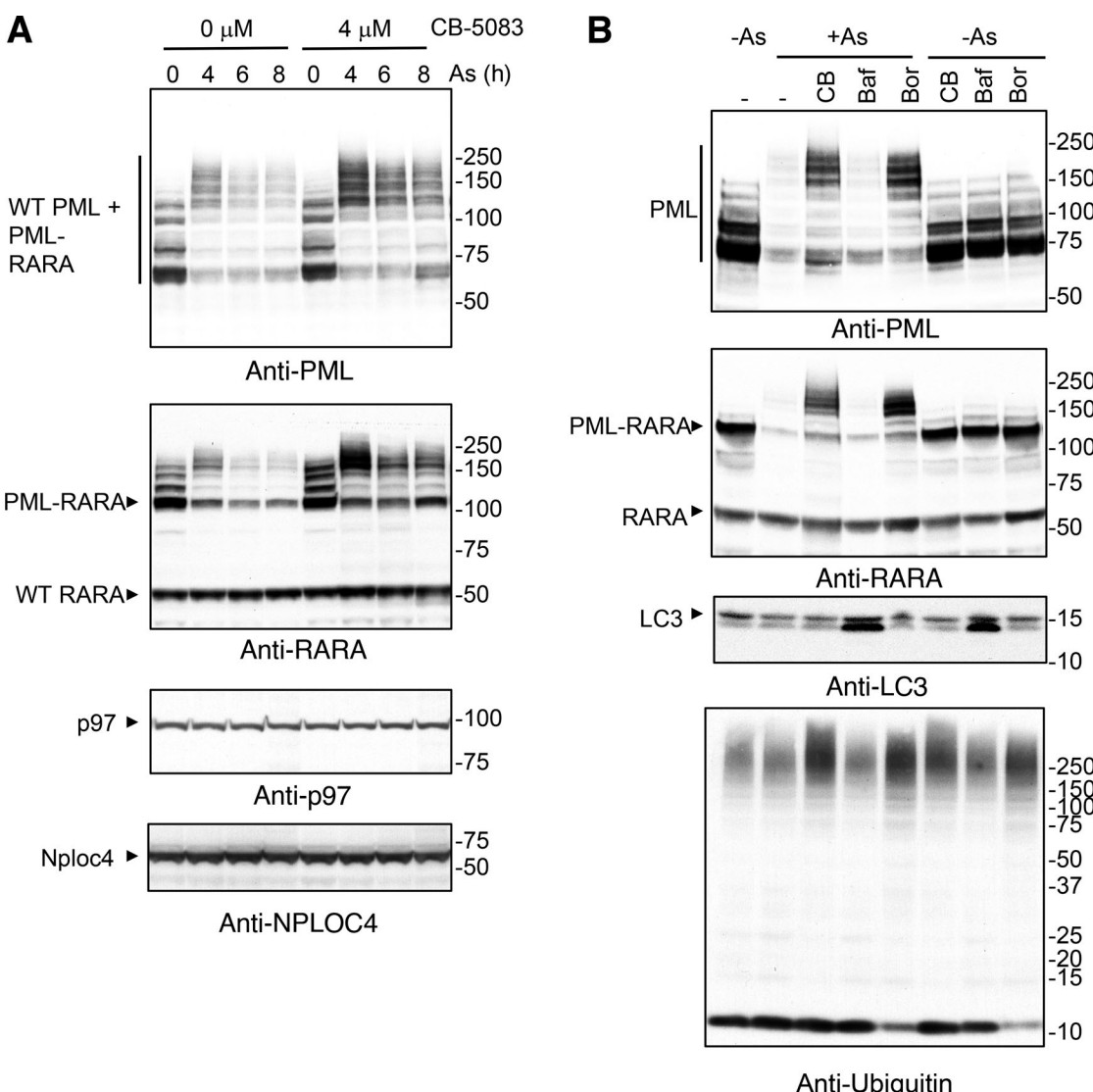

Figure 7. **Arsenic-induced degradation of the PML/RARA oncoprotein requires p97. (A)** NB4 cells were treated with the indicated concentrations of CB-5083 and exposed to arsenic (As) for the indicated time (h). Cell lysates were analyzed by Western blotting using an antibody to PML, RARA, p97, and NPLOC4. **(B)** NB4 cells were either untreated (−), treated with CB-5083 (CB), bafilomycin (Baf), or bortezomib (Bor). Response to arsenic (6 h) was investigated for all conditions. Cell lysates were analyzed by Western blotting using antibodies to PML, RARA, LC3, or ubiquitin. Molecular weight markers are indicated in kD. Source data are available for this figure: SourceData F7.

2008). Here, we showed that after ubiquitination of PML and PML-RARA, there is an additional step in the pathway in which the p97 segregase is recruited to PML bodies to extract PML and PML-RARA prior to proteasomal degradation. P97/VCP (or Cdc48 in yeast) is a highly expressed ATPase that in combination with its cofactors can extract proteins embedded in membranes or macromolecular structures, provided they are polyubiquitinated (Meyer and Weihl, 2014; Ye et al., 2017). Each subunit of the hexameric p97 is composed of an N-terminal (N) domain, two ATPase domains (D1 and D2), and a flexible C-terminal extension. The ATPase domains are arranged as two stacked rings enclosing a central channel with the D1 domains forming the upper ring and the D2 domains forming the lower ring. The biological activity of p97 is thus mediated by its ability to use the energy generated by ATP hydrolysis to mechanically

extract and unfold ubiquitinated proteins. P97 has been implicated in having a key role in many biological processes including ER-associated protein degradation, where misfolded and polyubiquitinated proteins are extracted from the ER and passed on to the proteasome for degradation (Christianson and Ye, 2014). P97 also functions in ribosome-associated quality control (Brandman and Hegde, 2016), the extraction of proteins from chromatin (Ramadan et al., 2007), and disassembly of the CMG helicase during DNA replication termination (Dewar and Walter, 2017; Maric et al., 2014; Moreno et al., 2014).

While p97 is the core of the force-generating machine, it is a range of 30 different cofactors that provide the specificity for p97 to function in such a diverse range of biological activities. The cofactors contain domains that recognize substrates, ubiquitin, and usually the N-domain of p97 (Stach and Freemont,

2017). An important cofactor is the UFD1/NPLOC4 hetrodimer that binds a K48-linked penta-ubiquitin chain. It is this cofactor that we have identified as being required for arsenic-induced degradation of PML and PML-RARA (Fig. 5), and this suggests that PML should be modified by such chains as a consequence of arsenic treatment. It is clear that ubiquitin accumulates on PML after arsenic treatment, and this is enhanced in the presence of the p97 inhibitor CB5083 (Fig. 6 D). Blotting with an antibody specific for K48-linked ubiquitin chains reveals that they are associated with PML, but only in the presence of the p97 inhibitor (Fig. 6 E). This suggests that these chains are a short-lived intermediate that under normal circumstances are rapidly extracted by p97 and degraded by the proteasome. As they are not completely removed from PML by treatment with a SUMO-specific protease, it is likely that a large proportion is directly attached to PML, rather than it all being linked to SUMO chains. Such a PML-linked K48 ubiquitin chain may represent the signal recognized by UFD1-NPLOC4 for p97 extraction, but at this stage, we cannot confirm that they reach the minimum length required for extraction. A requirement for a minimum of five K48-linked ubiquitins was demonstrated for p97-mediated extraction of the mcm7 component of the CMG helicase to terminate DNA replication (Deegan et al., 2020). An explanation for this requirement was provided by structural analysis of Cdc48-Ufd1-Npl4 unfolding an ubiquitinated substrate. The cofactors Ufd1 and Npl4 each bind two ubiquitin molecules with an additional ubiquitin trapped in an unfolded state within the central pore of Cdc48 (Twomey et al., 2019). It has previously been demonstrated that proteasomal subunits are recruited to PML bodies after arsenic treatment (Lallemand-Breitenbach et al., 2001). As p97 and ubiquitin also become associated with PML bodies after arsenic treatment, it suggests that p97 and the proteasome are in relatively close proximity.

Our observation that degradation of both PML and PML-RARA requires the p97 segregase suggested that both proteins are degraded via the same pathway. However, it has been suggested that ubiquitinated PML-RARA was degraded via autophagy (Isakson et al., 2010). Our studies indicate that in patient-derived NB4 cells, arsenic-induced degradation of both PML and PML-RARA is blocked by the proteasome inhibitor bortezomib, but not by the autophagy inhibitor bafilomycin (Fig. 7). Thus, it seems likely that after p97 extraction ubiquinated PML and PML-RARA are channeled to the proteasome for degradation.

PML and PML-RARA degradation in response to arsenic requires the activity of the STUbL RNF4 (Lallemand-Breitenbach et al., 2008; Tatham et al., 2008), which has the potential to not only generate ubiquitin chains, but also SUMO-ubiquitin hybrid chains. This SUMO–RNF4 axis has been shown to generate the ubiquitinated products required for p97 extraction of Fanconi anemia ID complex components in response to DNA damage (Gibbs-Seymour et al., 2015) and to remove cytotoxic trapped PARP1 from chromatin (Krastev et al., 2022). In both fission yeast and budding yeast, it has been demonstrated that Ufd1 not only has the ability to bind ubiquitin but also, via a SIM in its C-terminus, bind SUMO. This appears to facilitate the function of Ufd1 in STUbL mediated genome stability processes (Bergink

et al., 2013; Capella et al., 2021; Kohler et al., 2015; Nie et al., 2012; Nie and Boddy, 2016). While we cannot exclude the possibility that SUMO-ubiquitin hybrid chains play a role in arsenic-induced degradation of PML and PML-RARA, the SIM present in the C-terminus of yeast Ufd1 is not present in the human form of UFD1, suggesting that SUMO recognition by Ufd1 is not required in arsenic-induced p97 mediated extraction of PML from nuclear bodies. Thus, an updated mechanism (graphically illustrated in Fig. 8) for arsenic-induced degradation of PML and PML-RARA is that arsenic induces extensive modification of PML and PML-RARA by SUMO1 and SUMO2, which accumulate in nuclear bodies. The SUMO-targeted ubiquitin E3 ligase RNF4 is then recruited to the nuclear bodies where it ubiquitinates PML and PML-RARA. In turn, the ubiquitin signal recruits UFD1-NPLOC4-p97, which extracts the modified PML and PML-RARA from the nuclear bodies in a form that is suitable for degradation by the 26S proteasome.

## Materials and methods

### Preparation of anti-GFP nanobody magnetic beads
300 mg of Dynabeads M-270 Epoxy (Catalogue No. 14302D; Thermo Fisher Scientific) were resuspended in 10 ml of 0.1 M sodium phosphate buffer, pH 7.5, and washed twice using a DynaMag-15 magnet (Catalogue No. 12301D; Thermo Fisher Scientific) with 10 ml of the same buffer. 4 mg of the LaG16 nanobody that recognizes GFP (Fridy et al., 2014) in 10 ml of 1 M $(NH_4)_2SO_4$, 0.1 M sodium phosphate buffer, pH 8.0, was added to the beads, which were incubated overnight at 27°C with end-over-end rotation. After coupling was complete, the beads were washed six times with PBS and stored in PBS, 0.1% $NaN_3$ at 4°C.

### Purification of YFP-PML from U2OS PML$^{-/-}$ YFP-PML cells
For each "replicate" of the proteomics experiment, five 15-cm-diameter plates of ~80% confluent cells were used. Prior to harvesting, growth medium was removed and cells washed in situ twice with 5 ml PBS/100 mM iodoacetamide. For each dish, cells were scraped into 5 ml PBS+100 mM iodoacetamide and transferred to a 50-ml tube. A second 2 ml volume of PBS+100 mM iodoacetamide was used to clean the plate of all cells and pooled with the first 5 ml. This was repeated for all dishes for each replicate and pooled in the same 50-ml tube (~35 ml total). Cells were pelleted by centrifugation at 400 $g$ for 5 min at 22°C and resuspended in 10 ml 4°C hypotonic buffer including iodoacetamide (10 mM Hepes, pH 7.9, 1.5 mM MgCl2, 10 mM KCl, 0.08% NP-40, 1× EDTA-free protease inhibitor cocktail [Roche], + 100 mM iodoacetamide). A sample of this may be retained as a "whole-cell extract." These were snap-frozen in liquid nitrogen and stored at –80°C until required. Cells were thawed by tube rotation at 4°C for 30 min. Tubes were then centrifuged at 2,000 $g$, 4°C for 10 min to pellet nuclei. Supernatants were discarded, but a sample of this can be taken as a "cytosolic fraction." Each pellet of nuclei was resuspended in 5 ml ice-cold hypotonic buffer containing 20 mM DTT but without iodoacetamide, transferred to a 15-ml tube and centrifuged at 2,000 $g$, 4°C for 10 min. The nuclei were washed once more in 5 ml ice-cold hypotonic buffer (without iodoacetamide

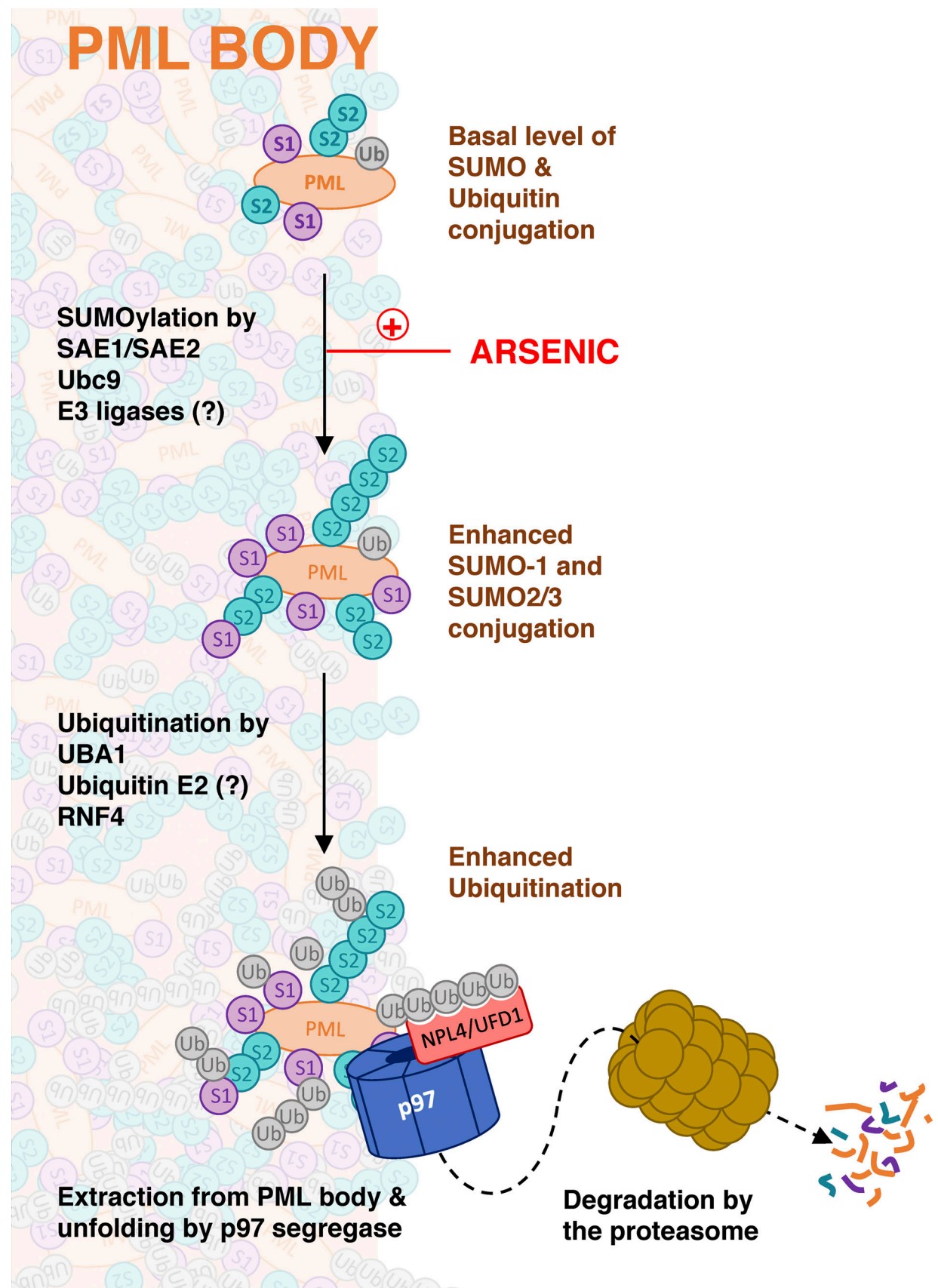

Figure 8. **Schematic illustrating a role for p97 in arsenic-induced degradation of PML.** Exposure of cells to arsenic triggers increased SUMO modification of PML. Multiple copies of SUMO recruit the ubiquitin E3 ligase RNF4, which synthesizes ubiquitin chains on PML. K48-linked ubiquitin chains are recognized by

the NPLOC4/UFD1 cofactors of the p97 segregase. P97 extracts and unfolds ubiquitinated and SUMOylated PML from PML bodies. PML is eventually degraded by the proteasome.

or DTT) and the supernatant discarded. To digest DNA, nuclei were resuspended in 5 ml "Buffer A" (50 mM Tris, pH 7.5, 150 mM NaCl, 0.2 mM DTT, 1 µg/ml benzonase, 1× EDTA-free protease inhibitor cocktail [Roche]) and incubated on ice for 15 min. Samples were sonicated in 15-ml tubes on ice (Branson Sonifier) using the small probe, six cycles of 30 s at ~50% amplitude with at least 2 min cooling on ice in between cycles. PML bodies should remain intact at this point. Insoluble debris was removed by centrifugation at 1,200 $g$ for 20 min at 4°C, and the supernatant containing soluble PML bodies, transferred to a new 15-ml centrifuge tube. A sample of this can be taken as a "nuclear fraction." To each 5 ml supernatant 6 ml "Buffer B" was added (50 mM Tris, pH 7.5, 1% Triton-X-100, 0.1% Deoxycholic acid, 1× EDTA-free protease inhibitor cocktail [Roche]). To each 11 ml solution, 1 ml volume of a 5% beads:buffer slurry (vol:vol) of magnetic anti-GFP nanobody beads, pre-equilibrated in 1:1.4 Buffer A:B mix (vol:vol), was added and mixed on a tube roller at 4°C for 16 h (50 µl beads per sample). Using the DynaMag rack (Thermo Fisher scientific), the beads were extracted from suspension and the supernatant removed. A sample of this can be analyzed to ensure YFP-PML depletion. Beads were then washed with 5 ml "Buffer C" (50 mM Tris, pH 7.5, 500 mM NaCl, 0.2 mM DTT) and the beads extracted. These beads were then resuspended in 1 ml "Buffer D" (50 mM Tris, pH 7.5, 150 mM NaCl, 0.2 mM DTT) and transferred to a new 1.5-ml protein LoBind tube (Eppendorf) and after bead separation washed once more with 1 ml Buffer D. One half of this final sample of beads (25 µl beads) was eluted for proteomic analysis by adding 35 µl 1.2X NuPAGE LDS sample buffer (Thermo Fisher Scientific), and incubated at 70°C for 15 min with agitation, followed by fractionation by SDS-PAGE. If Western blot analysis was also required, the second half of the resin (25 µl) would be resuspended in 250 µl 1.2X NuPAGE LDS sample buffer (Thermo Fisher scientific) and incubated at 70°C for 15 min with agitation. Approximately 30 µl would be used per gel lane. If treatment with proteases was required to monitor PML modification status, then the second half volume of beads (25 µl) was divided into four and treated as follows: (1) Add 150 µl Buffer D. (2) Add 150 µl 250 nM SENP1 in Buffer D. (3) Add 150 µl 500 nM USP2 in Buffer D. (4) Add 150 µl 250 nM SENP1/500 nM USP2 in Buffer D. All tubes were agitated for 1–2 h at 22°C, supernatants removed (and can be kept for analysis), and beads washed twice with 1 ml buffer D before elution with 250 µl 1.2X LDS, 70°C, 15 min with agitation.

## Proteomic analysis of YFP-PML purifications
Elutions from anti-GFP nanobody beads in 1.2× LDS/60 mM DTT protein sample buffer were fractionated on 4–12% polyacrylamide Bis-Tris NuPAGE gels (Thermo Fisher Scientific) using MOPS running buffer. After Coomassie staining and destaining gels were excised into four slices per lane and tryptic peptides extracted from each slice as previously described (Shevchenko et al., 2006). Peptides were resuspended in 40 µl 0.1% TFA 0.5% acetic acid and 6 µl of each analyzed by liquid chromatography–MS/MS. This was performed using a Q

Exactive mass spectrometer (Thermo Fisher Scientific) coupled to an EASY-nLC 1,000 liquid chromatography system (Thermo Fisher Scientific), using an EASY-Spray ion source (Thermo Fisher Scientific) running a 75 µm × 500 mm EASY-Spray column at 45°C. A 150 min elution gradient with a top 8 data-dependent method was applied. Full scan spectra (m/z 300–1,800) were acquired with resolution R = 70,000 at m/z 200 (after accumulation to a target value of 1,000,000 ions with maximum injection time of 20 ms). The eight most intense ions were fragmented by higher-energy collisional dissociation and measured with a resolution of R = 17,500 at m/z 200 (target value of 500,000 ions and maximum injection time of 120 ms) and intensity threshold of $2.1 \times 10^4$. Peptide match was set to "preferred," a 40 s dynamic exclusion list was applied, and ions were ignored if they had unassigned charge state 1, 8, or >8.

MS data were processed in MaxQuant version 1.6.1.0 (Cox and Mann, 2008; Cox et al., 2011). Only the top 3 slices were analyzed to avoid the highly abundant protein in the low molecular weight slice 4. Oxidized methionine and acetylated protein N-termini were selected as variable modifications and carbamidomethyl-C was the only fixed modification. Maximum number of variable modifications was set to 4. The match between runs option was enabled, which matches identified peaks among slices from the same position in the gel as well as one slice higher or lower. The uniport human proteome database (downloaded 19/4/2019; 73,920 entries) digested with Trypsin/P (maximum missed cleavages was 5) was used as search space. Label free quantitation (LFQ) intensities were required for each slice. All FDR filtering was set to 1%.

Data for 3,615 protein groups were returned, which was reduced to 2,157 by removal of decoy proteins, proteins only identified by modified peptides, proteins from the contaminants database, and proteins without a complete set of reported LFQ intensities for all four replicates of at least one experimental condition. LFQ intensities were further manually normalized as follows: For each protein detected in the same slice for all samples, the LFQ intensity relative to the average LFQ across all lanes was calculated. The median relative intensity for all proteins was used to normalize all protein LFQ values in that slice. The final protein LFQ intensity per lane (and therefore sample) was calculated by the sum of normalized LFQ values in all slices. Downstream data processing used Perseus v1.6.1.1 (Tyanova et al., 2016). Zero intensity values were replaced from $\log_2$ transformed data (width 0.3 and downshift 1.8). Outliers were defined using two-sample student's $t$ tests using significance FDR cutoffs of 10% (for + arsenic vs. untreated) and 0.05% (for WT U2OS vs. YFP-PML). S0 values were set at 0.2 for all. For volcano plots, fold changes were calculated by ratio of average LFQ intensities across all four replicates including zero replaced values.

## Pre-extraction of cells
Cells were grown on coverslips in 24-well plates. Coverslips were washed twice with PBS before being exposed to pre-

extraction buffer (25 mM Hepes, pH 7.4, 50 mM NaCl, 3 mM MgCl2, 0.5% Triton-X-100, 0.3 M Sucrose) for 10 min. Cells were then processed through the procedure below for immunofluorescence.

## Immunofluorescence analysis

NB4 suspension cells in exponential growth were added to poly-L-lysine coated coverslips in 24-well plates and the cells sedimented onto the coverslips by centrifugation at 400 × $g$ for 5 min. After treatment with arsenic and CB5083, cells were washed once with PBS, fixed in 2% paraformaldehyde in PBS, washed with PBS, and blocked in 3% IgG free BSA, 0.1% TWEEN 20 in PBS. Blocking buffer was removed and cells were incubated with primary antibody in blocking buffer for 3 h. In a humidified chamber. Primary antibodies were chicken anti-PML (in house) and rabbit anti-RARA (ab275748; Abcam). Cells were washed three times with 0.1% TWEEN 20 in PBS prior to incubation with secondary antibodies in blocking buffer as above. Secondary antibodies were Alexa Fluor 488 goat (Invitrogen), donkey anti-chicken IgG, and Alexa Fluor 594 goat anti-rabbit IgG. Cells were washed three times with 0.1% TWEEN 20 in PBS and counterstained with 0.1 µg/ml DAPI in the same buffer. Cells were subsequently embedded in Vectashield mounting medium, blotted dry, and stored at 4°C. Images were acquired on a restoration microscope (DeltaVision Spectris; Applied Precision) with a 100 × 1.40 NA objective and a cooled charge-coupled device camera (CoolSNAP HQ; Roper Scientific). Image analysis was performed by SoftWorx software (Applied Precision) and images were displayed using OMERO. Adherent cells were washed three times with PBS, fixed with 4% formaldehyde in PBS for 15 min, then washed again three times with PBS before blocking for 30 min in 5% BSA, 0.1% TWEEN 20 in PBS. Fixed cells were washed once with 1% BSA, 0.1% TWEEN 20 in PBS before incubation for 1–2 h with primary antibody in 1% BSA, 0.1% TWEEN 20 in PBS. These were chicken anti-PML (in house) and sheep anti-p97 (MRC Dundee). After three washes in 1% BSA, 0.1% TWEEN 20 in PBS, cells were incubated with secondary antibody in 1% BSA, 0.1% TWEEN 20 in PBS for 60 min, washed three times in 1% BSA, 0.1% TWEEN 20 in PBS, and stained with 0.1 µg/ml DAPI for 2 min. These were Alexa Fluor (Invitrogen) donkey anti-chicken 594, donkey anti-sheep 594, and donkey anti-chicken 488. Coverslips were rinsed twice with PBS, twice with water, and dried before mounting in MOWIOL. Images were collected using a DeltaVision DV3 widefield microscope and processed using SoftWorx (both Applied Precision). Images are presented as maximal intensity projections using OMERO software. For quantitative p97-PML colocalization analysis, the same image settings were applied across all fields and PML bodes per cell were counted for between 25 and 41 cells per condition. P97 colocalization was defined as any p97 signal above background within the same region as a PML body. Percent colocalization was then calculated for each cell.

## Live-cell imaging in real time and measurement of fluorescence intensities

For all live-cell imaging experiments, cells were seeded onto µ-slide 8-well glass bottom (80827; Ibidi). Immediately prior to imaging cell medium was replaced with DMEM minus Phenol Red (31053-038; Gibco) supplemented with 2 mM glutamine and 10% fetal bovine serum. To study the effects of p97 inhibition, cells were untreated, treated with either 1 µM arsenic, 5 µM CB-5083, or both for a total of 6.5 h. For siRNA analysis experiments, double-stranded siRNA were introduced at 10 nM final concentration using Lipofectamine RNAiMAX (Invitrogen). 2 d later, cells were either treated of not with 1 µM arsenic and monitored for 10 h. Time-lapse microscopy was performed on a DeltaVision Elite restoration microscope (Cytiva) fitted with an incubation chamber (Solent Scientific) set to 37°C and 5% $CO_2$ and a cooled charge-coupled device camera (CoolSnap HQ; Roper). SoftWorx software was used for image collection. Datasets were deconvolved using the constrained iterative algorithm (Swedlow et al., 1996; Wallace et al., 2001) using SoftWorx software. Using a YFP-specific filter set (Ex510/10 nm, Em537/26 nm), 5 z-planes spaced at 2 µm were taken every 15 min. A single brightfield reference image was taken at each timepoint to monitor cell health. Time courses were presented as maximum intensity projections of deconvolved 3D datasets. Movies were created in OMERO using the deconvolved images.

## Quantitative analysis of PML body composition during live-cell imaging

Batch analysis of time-lapse movies was carried out using ImageJ (Schneider et al., 2012) macros in Fiji (Schindelin et al., 2012; supplementary materials). Briefly: (1) datasets (deconvolved with SoftWorx) were cropped to remove border artifacts, (2) for each timepoint the maximum focus slice was selected according to intensity variance (radius 2 pixels), (3) auto-thresholding was performed using a threshold of mean intensity + 3 standard deviations, and (4) ImageJ's built-in "Analyze Particles" function was used to measure PML body count, size, total area, and intensity. Results were aggregated and summarized using R scripts (https://www.R-project.org; supplementary materials).

## siRNA-mediated depletion

Cells were transfected with a pool containing an equimolar amount of four siRNA duplexes essentially as described (Ibrahim et al., 2020) Dharmacon ON-TARGETplus; Human VCP (p97), Human UFD1, Human NPLOC4, Human UBXN6, Human FAF2, Human PLAA, Human RNF4, and Human SPRTN to a final concentration of 10 nM, or a non-targeting control duplex (siNT) at the same concentration using Lipofectamine RNAiMAX (Life Technologies) according to the manufacturer's instructions.

## Preparation of cell extracts and Western blotting

NB4 cells were collected by centrifugation, washed once with PBS, resuspended in PBS and made 1X LDS buffer (2% Lithium dodecyl sulfate, 250 mM Tris HCl, pH 8.4, 50 mM DTT, 0.5 mM EDTA, 10% glycerol, Coommassie Blue, and Phenol Red) by addition of 4X LDS buffer (Thermo Fisher Scientific) and heated to 70°C for 10 min. Adherent cells were washed with PBS and lysed by the addition of 1.2X LDS buffer and heated to 70°C for 10 min. Cell lysates were sonicated using a probe sonicator for two bursts of 30 s. Samples were run on NuPage 4–12% Bis-Tris precast gels (Thermo Fisher Scientific) and transferred to

nitrocellulose. Membranes were blocked in PBS with 5% non-fat milk and 0.1% Tween-20. Primary antibody incubations were performed in PBS with 3% BSA and 0.1% Tween-20. For the HRP-coupled secondary antibody incubations 5% milk was used instead of the BSA. The signal was detected by Pierce enhanced chemiluminescence (32106; Thermo Fisher scientific) and x-ray films. Antibodies: Chicken anti-PML (homemade), mouse anti-Lamin A/C (SAB4200236; Sigma-Aldrich), sheep anti-SUMO-1 (homemade), rabbit anti-SUMO-2 (4971S; Cell signaling), rabbit FK2 ubiquitin (Enzo BML-PW8810-0500), rabbit anti-p97 (ab109240; Abcam), sheep anti-UFD1 (MRC Dundee), sheep anti-NPLOC4 (MRC Dundee), rabbit anti-UBXN6 (ab221167; Abcam), rabbit anti-FAF2 (16251-1-AP; Proteintech), rabbit anti-PLAA (PA5-54299; Invitrogen), rabbit anti-RARA (ab275748; Abcam), and rabbit anti-LC3 A/B (4108S; Cell signaling). HRP-coupled secondary anti-mouse, anti-rabbit anti-chicken, and anti-sheep were purchased from Sigma-Aldrich.

### Statistical information

For live-cell imaging quantitation, 10 fields of view containing an average of 10 cells were monitored over the indicated time courses. For time-resolved data, the average PML body size, number of PML bodies per cell, total area of PML bodies, and total intensity of PML bodies were monitored in all 10 fields of view. For statistical comparisons among conditions at a fixed time-point, data were averaged by field of view, giving $n = 10$ for paired comparisons. These were made using Student's two-tailed unpaired $t$ tests with Welch's correction for unequal variances where necessary.

For colocalization studies using fixed cell immunofluorescence, the percentage of PML bodies colocalized with p97 was manually assessed on a per cell basis from randomly selected cells and two-tailed Mann-Whitney U-test used to make statistical comparisons between treated and untreated conditions. Cell counts; $n = 41$ (NT), $n = 38$ (2 h), $n = 38$ (4 h), and $n = 25$ (6 h).

For peptide intensity differences in proteomics data, Student's two-tailed unpaired $t$ tests were used with Welch's correction, where required. Peptide intensity values from four biological replicates were used.

Outliers in shotgun proteomics experiments were calculated in the Perseus statistical environment. These used two-sample Student's $t$ tests with FDR cutoffs of 10% (for + arsenic vs. untreated) and 0.05% (for WT U2OS vs. YFP-PML). S0 values were set at 0.2 for all.

All parametric tests assumed data to be normally distributed although this was not formally tested.

### Online supplemental material

Fig. S1 characterizes the U2OS PML$^{-/-}$ + YFP-PML cells by immunofluorescence and shows the response of YFP-PML to arsenic treatment by Western blot. Purified PML bodies are visualized by fluorescence microscopy and the purification of YFP-PML from cells monitored by Western blot. Fig. S2 shows a Coomassie stained SDS-PAGE gel of purified PML bodies used for proteomic analysis. It shows principal component analysis of overall proteomics data and statistical filtering of protein identifications to shortlist proteins specifically enriched in YFP-PML purifications and those proteins that change in abundance in purifications upon arsenic treatment. Colocalization of PML with MDN1 and FIB is investigated by fluorescence microscopy. The influence of SUMOylation and ubiquitination upon arsenic-induced PML/p97 colocalization was investigated using immunofluorescence after treatment with ML792 and TAK243 chemical inhibitors. The role of RNF4 and SPRTN was investigated in the same assay using siRNA treatment for each. Fig. S3 tests the involvement of the p97 cofactors UFD1, NPLOC4, UBXN6, FAF2, and PLAA on arsenic-induced PML degradation using siRNA ablation followed by Western-blot analysis. Fig. S4 monitors the influence of UFD1, NPLOC4, and UBXN6 on PML body area, count per cell, and size over a 10-h time course of arsenic treatment. Fig. S5 analyses the influence of CB5083 inhibition of p97 on arsenic-induced turnover of PML and PML-RAR in NB4 cells using fluorescence microscopy. Data S1 contains the processed MS proteomics data output files from MaxQuant. Videos 1, 2, 3, and 4 show changes to YFP-PML in U2OS PML$^{-/-}$ cells during exposure to arsenic or CB-5083 using fluorescence microscopy. Videos 5, 6, 7, 8, 9, and 10 show the effect of siRNA knock-down of selected p97 co-factors on the response of YFP-PML to arsenic exposure in U2OS PML$^{-/-}$ cells using fluorescence microscopy.

### Data availability

Immunofluorescence and Western blot data underlying all figures and supplementary figures are available in the source data file. The MS proteomics data have been deposited to the ProteomeXchange Consortium via the PRIDE partner repository (Perez-Riverol et al., 2019) with the dataset identifier PXD030330.

### Acknowledgments

We would like to thank Iain Porter of the Dundee Imaging Facility for invaluable help in setting up live-cell imaging experiments.

This work was supported by an Investigator Award from Wellcome Trust (217196/Z/19/Z) and a Program grant from Cancer Research UK (C434/A21747) to R.T. Hay. M. Liczmanska was supported by the European Union's Horizon 2020 research and innovation program under the Marie Sklodowska Curie grant agreement 765445 (UbiCode).

Author contributions: E.G. Jaffray generated cell lines, designed and performed most experiments, and interpreted data. A. Rojas-Fernandez generated PML expression constructs and genetically manipulated cell lines. Y. Yin carried out immunofluorescence analysis of NB4 cells. R.T. Hay carried out inhibitor analysis of NB4 cells and validated proteomics data. G. Ball developed the pipeline for the automated analysis of live-cell imaging experiments. M. Liczmanska carried out experiments on p97 recruitment. B. Mojsa carried out immunofluorescence experiments on requirements for p97 and nucleolar protein recruitment. M.H. Tatham consulted over proteomic experimental design, acquired and processed MS data, and conducted bioinformatic and statistical analyses. All authors contributed to data analysis. R.T. Hay, M.H. Tatham, and E.G. Jaffray wrote the paper. R.T. Hay conceived the project.

Disclosures: The authors declare no competing interests exist.

Submitted: 6 January 2022

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

**Supplemental material**

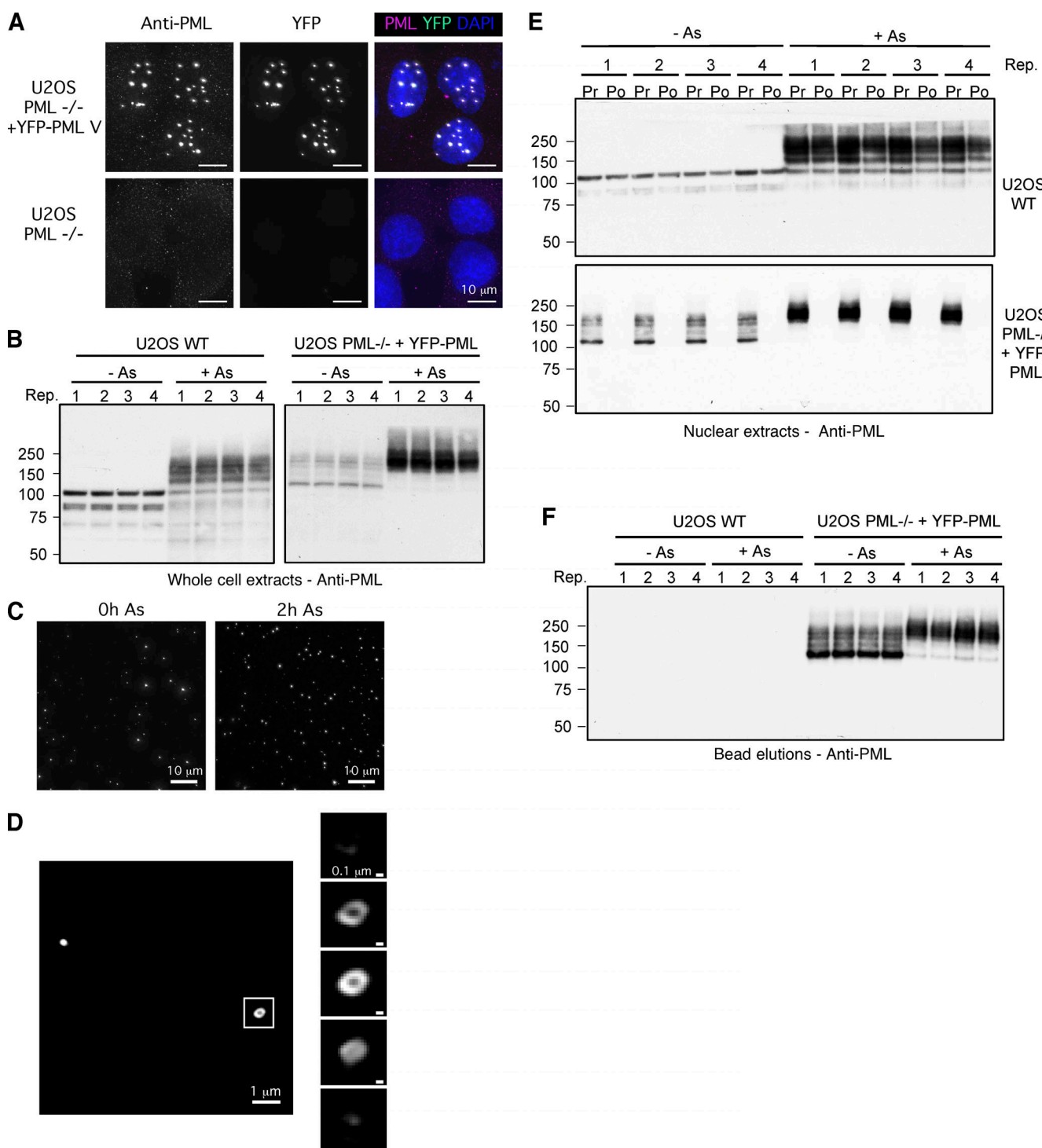

Figure S1. **Analysis of PML bodies purified from cultured cells. (A)** Fluorescence microscopy comparing U2OS PML$^{-/-}$ cells (lower) with the U2OS PML$^{-/-}$ + YFP-PML cells (upper). Images captured using chicken anti-PML with Texas Red secondary (left panels) or YFP fluorescence (center panels). Right panels show the two channels merged with DAPI. Scale bars are 10 μm. **(B)** U2OS WT and U2OS PML$^{-/-}$ + YFP-PML cells in quadruplicate were either untreated (−As) or treated with arsenic for 2 h (+As) and lysates analyzed by Western blotting with an antibody to PML. **(C)** Fluorescence microscopy of nuclear extracts from U2OS PML$^{-/-}$ + YFP-PML cells either untreated (−As) or treated with arsenic for 2 h (+As). **(D)** Left panels show a widefield and right panels show four z sections through a single PML body (untreated). **(E)** Anti-PML Western blots of nuclear extracts either prior (Pr) to or post (Po) incubation with anti-GFP nanobody beads. **(F)** Anti-PML Western blot of material eluted from anti-GFP nanobody beads. Molecular weight markers are indicated in kD. Source data are available for this figure: SourceData FS1.

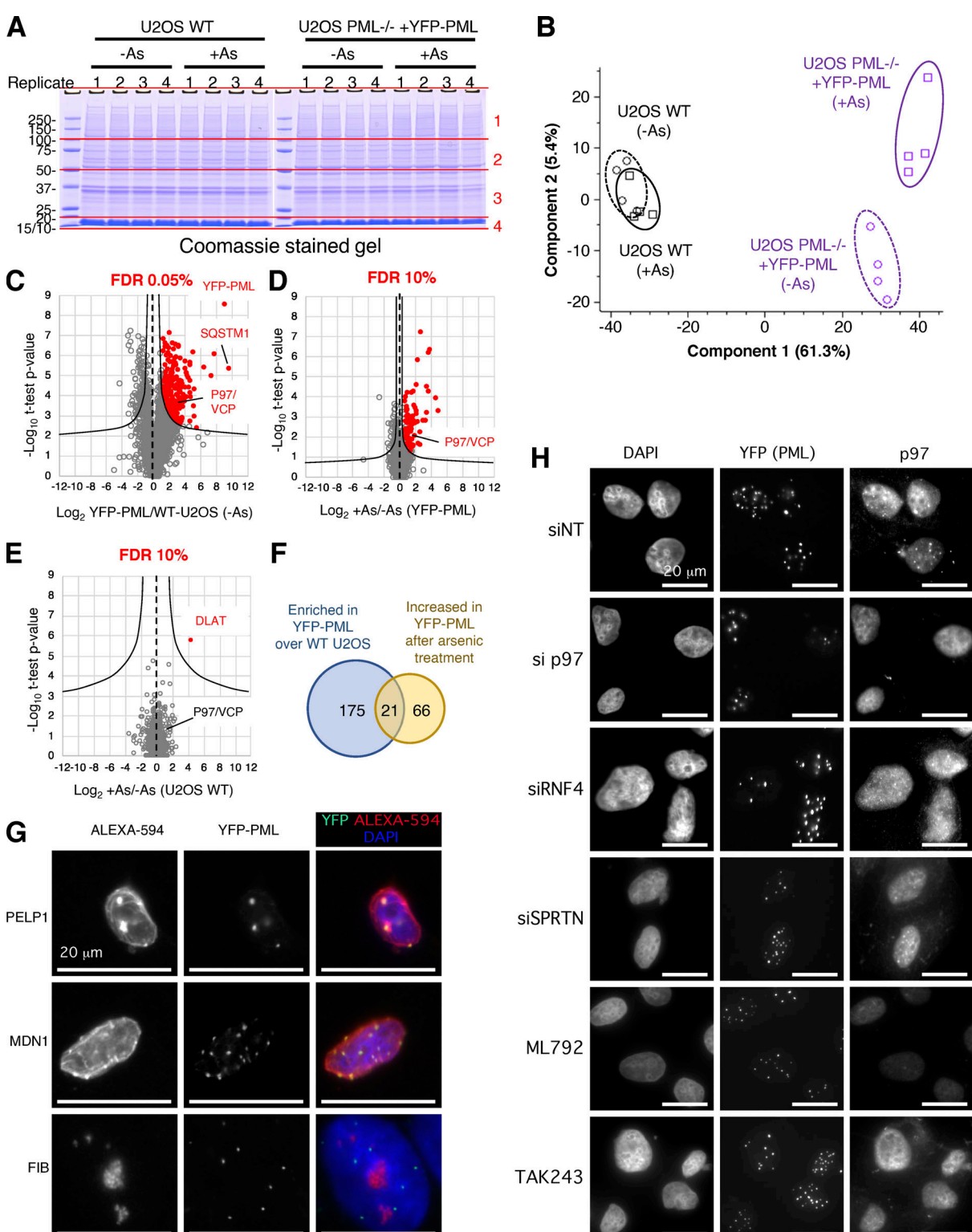

Figure S2. **VCP/p97 interaction with PML increases during arsenic treatment. (A)** Coomassie stained SDS-PAGE gel fractionating anti-GFP affinity purifications from the indicated cell types with or without arsenic treatment (As). Gels were sliced into four sections per lane (as indicated) for in-gel tryptic digestion. Molecular weight markers are indicated in kD. **(B)** Principal component analysis of intensity data for 2,156 identified proteins. **(C–E)** Scatter plots showing fold change versus two-tailed student's *t* test P value for the comparisons indicated on the x-axes (*n* = 4). FDR filtering values are indicated above each chart. **(F)** Venn diagram showing overlap between significant changers for parts C and D. The intersection is highlighted in Fig. 1 B. **(G)** Immunofluorescence analysis of PELP, MDN1, and FIB in U2OS PML$^{-/-}$ + YFP-PML using Alexa-594 secondary antibodies (left) and YFP fluorescence (center). Merged images with DAPI staining are shown (right panels). **(H)** In U2OS PML$^{-/-}$ + YFP-PML p97, RNF4 or SPRTN levels were ablated using siRNA or the SUMO and ubiquitin E1 enzymes inhibited with ML792 or TAK243, followed by 6 h treatment with arsenic. YFP-PML colocalization with p97 was measured by fluorescence and immunofluorescence. Source data are available for this figure: SourceData FS2.

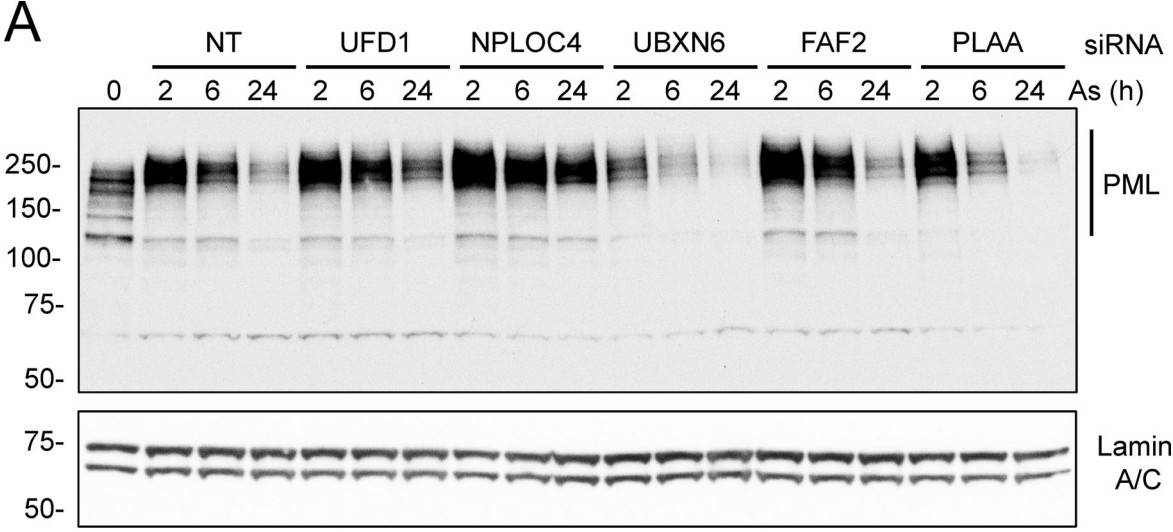

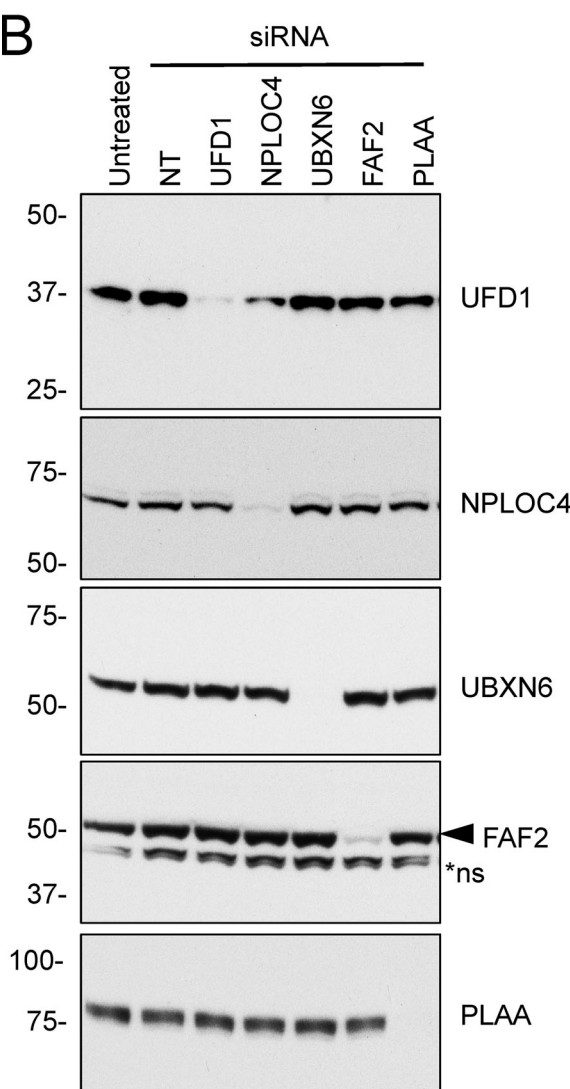

Figure S3. **Identification of UFD1 and NPLOC4 as the p97 cofactors involved in arsenic-induced PML degradation.** U2OS PML⁻/⁻ + YFP-PML cells were treated with NT or with siRNAs to the indicated p97 cofactors and exposed to arsenic for the times indicated. **(A)** Cell lysates were analyzed by Western blotting with antibodies to PML and Lamin A/C. **(B)** Western blotting for the indicated p97 cofactors in whole-cell extracts from cells treated with the indicated siRNAs. Molecular weight markers are indicated in kD. Source data are available for this figure: SourceData FS3.

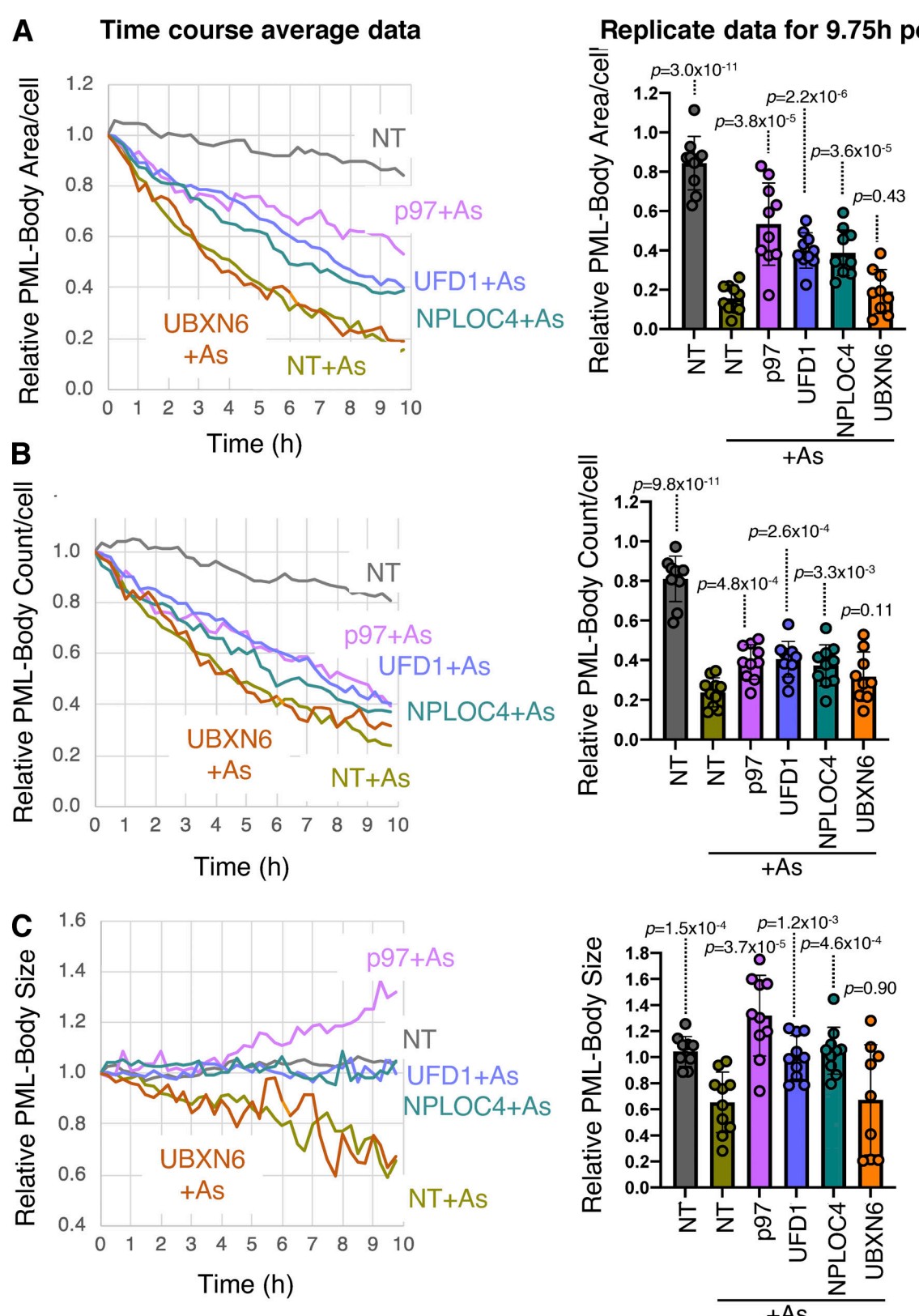

Figure S4. **UFD1 and NPLOC4 influence the nuclear distribution of PML.** U2OS PML$^{-/-}$ + YFP-PML cells were treated with NT siRNA or with siRNAs to the indicated p97 cofactors, exposed to arsenic and analyzed by time-lapse fluorescent imaging for 9.75 h. PML body parameters were plotted as a function of time (left panels; average—$n$ = 10). In the right panels PML body parameters are compared in cells treated with NT siRNA or with siRNAs to the indicated p97 cofactors after 9.75 h arsenic treatment (average value ± SD). P values are two-tailed Student's $t$ test assuming similar data deviations comparing with NT + As. Each data point is a single field of view taken from the time course ($n$ = 10). **(A)** Relative PML body area per cell. **(B)** Relative PML body number per cell. **(C)** Relative PML body size.

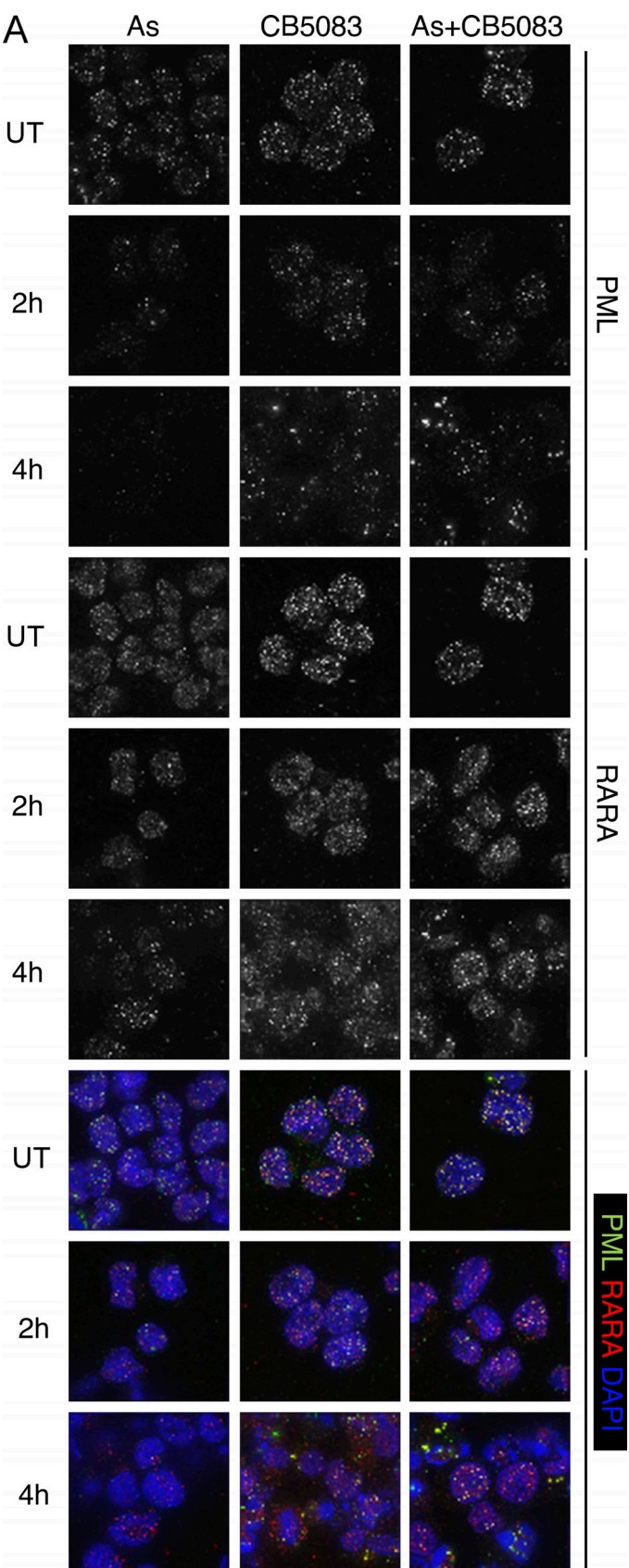

Figure S5. **Pharmacological inhibition of p97 interferes with arsenic-induced degradation of PML and PML-RARA.** NB4 cells were either untreated (UT) or treated with arsenic (As), CB-5083, or a combination of arsenic and CB-5083 for the times indicated. Cells were centrifuged onto coverslips, fixed and stained with antibodies to PML (green) and RARA (red), and DNA visualized by DAPI staining (blue).

Video 1 **No treatment.** Associated with Fig. 3 A. Untreated cells: U2OS PML$^{-/-}$ + YFP-PMLV cells, monitored by fluorescence microscopy every 15 min for 6.75 h (6.75 fps). 5 z-planes spaced at 2 µm. DeltaVision Elite restoration microscope with incubation chamber set to 37°C and 5% $CO_2$. Green indicates YFP (filter set Ex510/10 nm, Em537/26 nm). Scale bars are 5 µm.

Video 2 **AsO3.** Associated with Fig. 3 A. 1 µM arsenic: U2OS PML$^{-/-}$ + YFP-PMLV cells, monitored by fluorescence microscopy every 15 min for 6.75 h (6.75 fps). 5 z-planes spaced at 2 µm. DeltaVision Elite restoration microscope with incubation chamber set to 37°C and 5% $CO_2$. Green indicates YFP (filter set Ex510/10 nm, Em537/26 nm). Scale bars are 5 µm.

Video 3 **CB-5083.** Associated with Fig. 3 A. 5 µM CB-5083: U2OS PML$^{-/-}$ + YFP-PMLV cells, monitored by fluorescence microscopy every 15 min for 6.75 h (6.75 fps). 5 z-planes spaced at 2 µm. DeltaVision Elite restoration microscope with incubation chamber set to 37°C and 5% $CO_2$. Green indicates YFP (filter set Ex510/10 nm, Em537/26 nm). Scale bars are 5 µm.

Video 4 **AsO3+CB-5083.** Associated with Fig. 3 A. 1 µM arsenic + 5 µM CB-5083: U2OS PML$^{-/-}$ + YFP-PMLV cells, monitored by fluorescence microscopy every 15 min for 6.75 h (6.75 fps). 5 z-planes spaced at 2 µm. DeltaVision Elite restoration microscope with incubation chamber set to 37°C and 5% $CO_2$. Green indicates YFP (filter set Ex510/10 nm, Em537/26 nm). Scale bars are 5 µm.

Video 5 **No treatment.** Associated with Fig. 5 B. Non-targeting SiRNA, no treatment: U2OS PML$^{-/-}$ + YFP-PMLV cells, monitored by fluorescence microscopy every 15 min for 9.75 (9.75 fps). 5 z-planes spaced at 2 µm. DeltaVision Elite restoration microscope with incubation chamber set to 37°C and 5% $CO_2$. Green indicates YFP (filter set Ex510/10 nm, Em537/26 nm). Scale bars are 5 µm.

Video 6 **NT + AsO3.** Associated with Fig. 5 B. NT SiRNA + 1 µM arsenic: U2OS PML$^{-/-}$ + YFP-PMLV cells, monitored by fluorescence microscopy every 15 min for 9.75 (9.75 fps). 5 z-planes spaced at 2 µm. DeltaVision Elite restoration microscope with incubation chamber set to 37°C and 5% $CO_2$. Green indicates YFP (filter set Ex510/10 nm, Em537/26 nm). Scale bars are 5 µm.

Video 7 **Sip97+AsO3.** Associated with Fig. 5 B. SiRNA to p97 + 1 µM arsenic: U2OS PML$^{-/-}$ + YFP-PMLV cells, monitored by fluorescence microscopy every 15 min for 9.75 (9.75 fps). 5 z-planes spaced at 2 µm. DeltaVision Elite restoration microscope with incubation chamber set to 37°C and 5% $CO_2$. Green indicates YFP (filter set Ex510/10 nm, Em537/26 nm). Scale bars are 5 µm.

Video 8 **SiUFD1+AsO3.** Associated with Fig. 5 B. SiRNA to UFD1 + 1 µM arsenic: U2OS PML$^{-/-}$ + YFP-PMLV cells, monitored by fluorescence microscopy every 15 min for 9.75 (9.75 fps). 5 z-planes spaced at 2 µm. DeltaVision Elite restoration microscope with incubation chamber set to 37°C and 5% $CO_2$. Green indicates YFP (filter set Ex510/10 nm, Em537/26 nm). Scale bars are 5 µm.

Video 9 **SiNPL4+AsO3.** Associated with Fig. 5 B. SiRNA to NPL4 + 1 µM arsenic: U2OS PML$^{-/-}$ + YFP-PMLV cells, monitored by fluorescence microscopy every 15 min for 9.75 (9.75 fps). 5 z-planes spaced at 2 µm. DeltaVision Elite restoration microscope with incubation chamber set to 37°C and 5% $CO_2$. Green indicates YFP (filter set Ex510/10 nm, Em537/26 nm). Scale bars are 5 µm.

Video 10 **SiUBXN6+AsO3.** Associated with Fig. 5 B. SiRNA to UBXN6 + 1 µM arsenic: U2OS PML$^{-/-}$ + YFP-PMLV cells, monitored by fluorescence microscopy every 15 min for 9.75 (9.75 fps). 5 z-planes spaced at 2 µm. DeltaVision Elite restoration microscope with incubation chamber set to 37°C and 5% $CO_2$. Green indicates YFP (filter set Ex510/10 nm, Em537/26 nm). Scale bars are 5 µm.

**Provided online is Data S1, which shows T-test significant YFP-PML+ATO/WT U2OS+ATO.**

