## [Peer Review File · The Journal of Cell Biology]

The p97/VCP segregase is essential for arsenic induced degradation of PML and PML-RARA

Ellis Jaffray, Michael Tatham, Barbara Mojsa, Magda Liczmanska, Alejandro Rojas-Fernandez, Yili Yin, Graeme Ball, and Ronald Hay

Corresponding Author(s): Ronald Hay, University of Dundee

Review Timeline:

Submission Date:	2022-01-06
Editorial Decision:	2022-02-11
Revision Received:	2022-10-27
Editorial Decision:	2022-12-21
Revision Received:	2022-12-23

Monitoring Editor: Pier Paolo Di Fiore

Scientific Editor: Lucia Morgado-Palacin

Transaction Report:

DOI: <https://doi.org/10.1083/jcb.202201027>

February 11, 2022

Re: JCB manuscript #202201027

Dr. Ronald T Hay
University of Dundee
Centre for Gene Regulation and Expression Sir James Black Centre Dow Street
Dundee, Scotland DD15EH
United Kingdom

Dear Dr. Hay,

Thank you for submitting your manuscript entitled "The p97/VCP segregase is required for arsenic induced degradation of PML and the PML-RARA oncoprotein". The manuscript was assessed by expert reviewers, whose comments are appended to this letter. We invite you to submit a revision if you can address the reviewers' key concerns, as outlined here.

You will see that reviewers are enthusiastic about the work and find the data convincing. However, they concur with the need of strengthening the mechanistic data and firmly demonstrating the relevance of the pathway, thus addressing reviewers' concerns with respect to these points would be required (rev #1 points 1-2 and point 5 & rev #3 point 2 and point 6). We agree with reviewers that validation of the proteome analyses is necessary (rev #1 point 1 & rev #2 p2-3). Nonetheless, in our view, repeating the mass spectrometry studies with endogenous PML is not needed to support the conclusions so addressing reviewer #2's point 4 is not a pre-requisite for publication. This reviewer also requests the use of primary APL cells for several experiments (points 1, 7 and 8). Of course, these experiments would enhance the value of the paper, but we understand that these are difficult models to obtain and to handle. Given that the study under consideration does not claim medical relevance or and/or therapeutic possibilities, we think the performance of these experiments is outside the major scope of the study; thus, addressing these points would not be strictly required for resubmission. We also think that answering reviewer #1's point 3 and reviewer #3's points 3-4 would increase the impact of the study; however, we recognize these can be experiments for follow-up studies. Therefore, they would not be required for publication, although the authors are welcome to include them if they have already initiated experiments in this line. We would expect that all other referee points are addressed.

GENERAL GUIDELINES:

Text limits: Character count for an Article is < 40,000, not including spaces. Count includes title page, abstract, introduction, results, discussion, acknowledgments, and figure legends. Count does not include materials and methods, references, tables, or supplemental legends.

Figures: Articles may have up to 10 main text figures. Figures must be prepared according to the policies outlined in our Instructions to Authors, under Data Presentation, <https://jcb.rupress.org/site/misc/ifora.xhtml>. All figures in accepted manuscripts will be screened prior to publication.

Supplemental information: There are strict limits on the allowable amount of supplemental data. Articles may have up to 5 supplemental figures. Up to 10 supplemental videos or flash animations are allowed. A summary of all supplemental material should appear at the end of the Materials and methods section.

Please note that JCB now requires authors to submit Source Data used to generate figures containing gels and Western blots with all revised manuscripts. This Source Data consists of fully uncropped and unprocessed images for each gel/blot displayed in the main and supplemental figures. Since your paper includes cropped gel and/or blot images, please be sure to provide one Source Data file for each figure that contains gels and/or blots along with your revised manuscript files. File names for Source Data figures should be alphanumeric without any spaces or special characters (i.e., SourceDataF#, where F# refers to the associated main figure number or SourceDataFS# for those associated with Supplementary figures). The lanes of the gels/blots should be labeled as they are in the associated figure, the place where cropping was applied should be marked (with a box), and molecular weight/size standards should be labeled wherever possible.

As you may know, the typical timeframe for revisions is three to four months. However, we at JCB realize that the implementation of social distancing and shelter in place measures that limit spread of COVID-19 also pose challenges to scientific researchers. Lab closures especially are preventing scientists from conducting experiments to further their research. Therefore, JCB has waived the revision time limit. We recommend that you reach out to the editors once your lab has reopened to decide on an appropriate time frame for resubmission. Please note that papers are generally considered through only one revision cycle, so any revised manuscript will likely be either accepted or rejected.

Thank you for this interesting contribution to Journal of Cell Biology. You can contact us at the journal office with any questions, cellbio@rockefeller.edu.

Sincerely,

Pier Paolo Di Fiore, MD, PhD
Editor
The Journal of Cell Biology

Lucia Morgado-Palacin, PhD
Scientific Editor
Journal of Cell Biology

Reviewer #1 (Comments to the Authors (Required)):

In acute promyelocytic leukemia (APL), a translocation leads to fusion of PML with RARA causing transcriptional reprogramming that inhibits differentiation and stimulates proliferation. APL is treated with arsenic that induces degradation of PML-RARA which in turn attenuates proliferation. The authors have performed sophisticated mass spectrometry on PML bodies before or after treatment and identified p97. They demonstrate in great detail that PML (or PML-RARA) is first SUMOylated, followed by RNF4-mediated ubiquitylation, and then targeted by p97 to facilitate the degradation in the proteasome.

The data is of high quality and convincing. The study deciphers an important pathway that mediates the cellular effects and response to APL treatment and implicates p97, which itself is target of cancer therapy development. In doing so, the study also establishes key aspects of cellular regulation by SUMO and ubiquitin, RNF4 and p97. As outlined below, I have only few technical points. However, the relevance of this pathway for therapy response should be addressed in APL cells, which should be easy. Moreover, the cooperation of RNF4 and p97 should be demonstrated better as PML appears to be a good model for that in the mammalian system, and this lies within the authors expertise.

- 1) The authors clarify the mechanistic hallmarks in U2OS osteosarcoma cells as a convenient model. They also confirm a few aspects in the NB4 cells (Fig. 7), which is important because NB4 cells are a patient derived APL cell line that is driven by PML-RARA. To further prove medical relevance of their findings, however, the authors should clarify in these cells whether elements of their degradation pathway are critical for the behavior and therapy response i.e. would RNF4 or p97 inactivation prevent differentiation and affect survival/proliferation of NB4 upon arsenic treatment.
- 2) Contrary to the situation in yeast, mammalian p97 and the Ufd1-Npl4 adapter do not seem to target SUMO directly, which is an important point for many nuclear functions of p97. The implication is that targeting of PML by p97 highly depends on RNF4. The authors should demonstrate that in order to support their model and resolve this question.
- 3) Given the segregase function of p97, the question arises what PML is segregated from. What is the binding partner? Are PML bodies resolved upon removal of PML? Is there information in the MS data?
- 4) Fig. 4A raises some doubts. First, PML bodies should be degraded upon arsenic treatment according to Fig. 3A, which is not seen here. Moreover, the p97 antibody staining was performed after pre-extraction of the cells. Controls should be provided for

specificity such as p97 depletion, or results confirmed with tagged p97. Is p97 translocation RNF4 and PML dependent?

5) The authors should make use of the p97 Walker B substrate trapping mutant that would support the implication of PML as substrate of p97. Can the mutant be specifically co-immunoprecipitated with PML compared to wild type p97?

Minor points:

6) Labelling is done inconsistently with ATO, As, arsenic. The abbreviation ATO is not explained.

7) lane 498-504: The authors elaborate on colocalisation of proteasome and p97 supporting cooperation. This seems unnecessary and inconsistent with ERAD where p97 and the proteasome segregate in different domains, shown by the Baumeister group. Also, the cited paper by Andreas Martin's group does support proximity between proteasome and p97.

8) The movies do not need to be published online as such.

Reviewer #2 (Comments to the Authors (Required)):

This work aims to show that the p97/VCP segregase is required to extract SUMOylated and ubiquitinated PML from PML bodies prior to degradation by the proteasome.

Main points

1. The majority of the experiments have been carried out in a single cell model (or derivatives of). More models (including primary APLs) are needed to give mechanistic proof of the authors hypothesis.
2. IP experiments should be used to corroborate proteome analyses.
3. Proteome analyses both in U2OS as well as in APL cell based model (NB4) should be performed in presence of CB-5083.
4. The experiments shown in Fig 1 with exogenous proteins, should also be demonstrated with endogenous PML and also in the APL context (NB4) with PML-RARa with and without Arsenic.
5. The p97/VCP segregase RNA levels should be also shown in all cell models used in Fig 1.
6. The relevance of p97/VCP segregase should be corroborated by applying Crisp-cas or siRNA targeting and verifying the effects in presence or absence of arsenic, CB-5083.
7. The pharmacological inhibition with CB-5083 of p97/VCP segregase should be also shown in different cell models such as primary APL.
8. Similarly the evaluation of the action of CB-5083 on NBs should be performed also in primary APL, APL models as well as in different models which are less artificial.
9. p97 cofactors UFD1 and NPLOC4 which seem to be required for arsenic induced degradation of PML are in the proteome analyses performed?

Minor points

All raw data should be made available.

Reviewer #3 (Comments to the Authors (Required)):

PML and the oncogenic fusion protein PML-RAR are considered as model substrates for SUMO-dependent ubiquitylation by the SUMO-targeted ubiquitin ligase (StUbl) RNF4. Ubiquitylation and proteasomal degradation of PML and PML-RAR can be triggered by arsenic trioxide, which induces SUMO chain formation on both proteins. Here, the authors demonstrate that arsenic-induced degradation of PML and PML-RAR depends on the AAA ATPase p97/VCP and its cofactors UFD1-NPLOC4. The work was initiated by a proteomic study of purified PML nuclear bodies, in which p97 was detected as a PML associated protein, whose binding was induced upon short term treatment of cells with arsenic. This finding could be validated by immunofluorescence experiments. Further data demonstrate that the p97 inhibitor CB-5083 prevents arsenic-induced degradation of PML and PML-RAR and alters the size and morphology of PML NBs. A systematic exploration of p97 co-factors further indicate that UFD1 and NPLOC4 are involved in these processes.

Altogether, the data provide convincing evidence that p97-UFD1-NPLOC4 are required for the degradation of PML and PML-RAR most likely by extracting the ubiquitylated species from PML bodies. These findings are in line with previous and very recent reports describing an involvement of p97 in extracting mammalian SUMOylated/ubiquitylated StUbl targets, such as FANCI or PARP1, from chromatin (Gibbs-Seymour et al., Mol Cell, 2015; Krastev et al., Nature Cell Biology 2022). Similar observations have also been made on the p97 orthologue CDC48 in lower eukaryotes, such as fission and baker's yeast. While this takes away some originality, the finding that p97 is involved in extracting proteins also from PML bodies and is required for PML degradation is novel and of considerable significance. Overall, the data are mostly convincing and support the conclusions.

Nevertheless, I do have a few comments and suggestions that might further strengthen the work.

Major points:

1) As mentioned above, the work was initiated by a proteomic study of purified PML nuclear bodies under control conditions or upon treatment of cells with arsenic trioxide. Purification of intact PML NBs has been a major challenge due to the insolubility of these structure. Only very recently, proximity-based proteomics have facilitated the identification of PML body components in a system-wide manner (Barroso-Gomila, Nature communications, 2021). In the current work the authors performed a biochemical purification of PML bodies, which is based on subcellular fractionation and final affinity purification on anti GFP nanobody beads using YFP-PML as a bait. Using this approach, the authors identified around 200 PML associated proteins under basal conditions and defined a set of 21 proteins, whose association with PML bodies is enhanced upon arsenic treatment (Figure 1C). This set contains p97/VCP as well as a number of bona fide PML body components, such as Topors, PIAS1/3, SUMO etc. Most of the proteins are annotated as NB-resident proteins validating the approach. Furthermore, there is also a fairly good overlap with the data obtained by proximity-based proteomics. Interestingly, however, the authors also identified a set of nucleolar proteins, such as MDN1, PELP1 or SIMC1, which were so far not reported to reside in PML bodies. It would be important to validate that these proteins are indeed associated with PML bodies under basal conditions and/or upon treatment with arsenic. To my opinion, this is important to exclude that the purification is partially contaminated with nucleoli. Alternatively, SIMC1 or MDN1 may associate with SUMOylated PML only after cell lysis due to the presence of characterized SUMO binding motifs in both proteins.

Even though the focus of the work is on p97 rather than any of the other identified proteins, this is not relevant for any of the follow-up experiments on p97. Nevertheless, this point should be addressed or a post-lysis artefact should at least be discussed.

2) While ubiquitin binding of the mammalian p97-UFD1-NPLOC4 complex is quite well understood, it is still not entirely clear whether SUMOylation of substrates or hybrid SUMO-UB chains can promote binding. To see whether ubiquitylation is strictly required for recruitment of p97 to PML bodies, cells could be co-treated with arsenic and an Ub E1 inhibitor, such as TAK-243.

3) Gibbs-Seymour et al. (Mol Cell, 2015) reported that p97 cooperates with the cofactor DVC1/SPRTN in the extraction of SUMOylated/Ubiquitylated FANCI/FANCD2 from chromatin. Based on these findings it would be interesting to see whether and how depletion of DVC1/SPRTN affect PML degradation and PML morphology. This could be particularly interesting in case K63 chains are detected on PML (see below).

4) As shown previously, endogenous PML (or at least some isoforms) are relatively resistant to arsenic induced degradation (also when compared to PML-RAR) under conditions of endogenous RNF4 expression (Figure 2D). Nevertheless, the pattern of higher molecular weight-bands is indicative of SUMOylation-ubiquitylation. This raises the question whether some ubiquitylated species on PML are non-proteolytic. To address this issue, immunoblotting in Figure 6D should be done with Ub K63/K48 linkage-specific antibodies in addition to the pan-Ub antibody FK2.

5) Gibbs-Seymour et al., Mol Cell, 2015; Krastev et al., Nature Cell Biology 2022 should be cited.

Dear Dr Fiore

JCB Manuscript - Editorial Decision 202201027

Please find attached our revised manuscript. Also attached is our response to the reviewers. As was previously agreed the requests by Reviewer 2 to carry out experiments in primary cells and to repeat the proteomic experiments in multiple different cell types was felt to be outwith the scope of the manuscript. Responses to the comments of Reviewer 2 are therefore rather cursory. Based on the constructive comments of reviewers 1 and 3 we have carried out additional experiments and reconfigured the Figures as follows.

1. We have added the mass spec validation pulldown data to Fig. 1.
2. To reduce the number of Supplementary Figures to 5 we have removed Supplementary Figs 3, 6 and 8, but have taken the ubiquitin gel from the original Supplementary Fig 8 and added this to Fig. 7B.
3. We have added Western blot data with ubiquitin K48 and K63 chain specific antibodies to Fig. 6. We have also added mass spec data on the K48 and K63 chains to the bottom of Fig. 6
4. We have added immunofluorescence data on the requirements for p97 recruitment into PML bodies to Supplementary Fig. 2.
5. We have added immunofluorescence data on the association of nucleolar proteins PELP1 and MDN1 with PML bodies to Supplementary Fig. 2.
6. Pulldown data with the Walker B p97 mutants has been added to Fig. 4.

A "clean" version of the revised paper is attached, as it the original version with all changes indicated.

Best wishes

Ron

Responses to comments are in **RED**.

Reviewer #1 (Comments to the Authors (Required)):

In acute promyelocytic leukemia (APL), a translocation leads to fusion of PML with RARA causing transcriptional reprogramming that inhibits differentiation and stimulates proliferation. APL is treated with arsenic that induces degradation of PML-RARA which in turn attenuates proliferation. The authors have performed sophisticated mass spectrometry on PML bodies before or after treatment and identified p97. They demonstrate in great detail that PML (or PML-RARA) is first SUMOylated, followed by RNF4-mediated ubiquitylation, and then targeted by p97 to facilitate the degradation in the proteasome.

The data is of high quality and convincing. The study deciphers an important pathway that mediates the cellular effects and response to APL treatment and implicates p97, which itself is target of cancer therapy development. In doing so, the study also establishes key aspects of cellular regulation by SUMO and ubiquitin, RNF4 and p97. As outlined below, I have only few technical points. However, the relevance of this pathway for therapy response should be addressed in APL cells, which should be easy. Moreover, the cooperation of RNF4 and p97 should be demonstrated better as PML appears to be a good model for that in the mammalian system, and this lies within the authors expertise.

1) The authors clarify the mechanistic hallmarks in U2OS osteosarcoma cells as a convenient model. They also confirm a few aspects in the NB4 cells (Fig. 7), which is important because NB4 cells are a patient derived APL cell line that is driven by PML-RARA. To further prove medical relevance of their findings, however, the authors should clarify in these cells whether elements of their degradation pathway are critical for the behavior and therapy response i.e. would RNF4 or p97 inactivation prevent differentiation and affect survival/proliferation of NB4 upon arsenic treatment.

This is a good idea and the best cells to do this experiment are the patient derived NB4 cells that we have used in the paper. However the difficulty is that the arsenic induced differentiation experiment takes 5 days to perform (Misra et al., Selenite promotes all-trans retinoic acid-induced maturation of acute promyelocytic leukemia cells. *Oncotarget*. 2016 7:74686-74700). Over this time scale we find that siRNA to RNF4 is not effective and we have been unsuccessful in generating a CRISPR/cas9 mediated knockout of RNF4. The other suggestion is to inactivate p97, but p97 expression is required for long-term viability and we find that either pharmacological inhibition or knockdown of p97 results in cell death before differentiation can be observed. It has been demonstrated that cancer cells are uniquely sensitive to p97 inhibition and this forms the basis of clinical trials testing p97 inhibitors for cancer therapy (Anderson et al., Targeting the AAA ATPase p97 as an Approach to Treat Cancer through Disruption of Protein Homeostasis. *Cancer Cell*. 2015; 28:653-665; Le Moigne et al., The p97 Inhibitor CB-5083 Is a Unique Disrupter of Protein Homeostasis in Models of Multiple Myeloma. *Mol Cancer Ther*. 2017 16:2375-2386). We also tried to block ubiquitination with the ubiquitin E1 Activating enzyme inhibitor TAK243 (Hyer et al., A small-molecule inhibitor of the ubiquitin activating enzyme for cancer treatment. *Nat Med*. 2018; 24:186-193), but again this resulted in cell death before we could evaluate differentiation. We were thus unable to perform the differentiation experiment. The long-term toxicity of p97 inhibition is also why we restricted our treatment of cells with CB-5083 to 8 hours or less. Over this time period we did not observe any adverse effects on the cells.

2) Contrary to the situation in yeast, mammalian p97 and the Ufd1-Npl4 adapter do not seem to target SUMO directly, which is an important point for many nuclear functions of p97. The implication is that targeting of PML by p97 highly depends on RNF4. The authors should demonstrate that in order to support their model and resolve this question.

This has been done (see Supplementary Fig 2H).

3) Given the segregase function of p97, the question arises what PML is segregated from. What is the binding partner? Are PML bodies resolved upon removal of PML? Is there information in the MS data?

PML bodies are scaffolded by PML itself and do not require SUMO modification for their assembly. However many of the proteins that associate with PML bodies (identified in our proteomic analysis) do so via SUMO-SIM interactions and inhibition of SUMO modification causes their release while leaving PML bodies intact. Thus we feel that PML is probably being segregated from PML itself. PML forms dimers via its long coiled coil but also appears to multimerise through homotypic interactions of both B-box 1 and B-box 2. Thus, PML bodies are held together by multiple PML-PML interactions that would need to be broken to extract PML monomers for proteasomal degradation. It does appear that PML bodies are being resolved as PML body area per cell and PML body numbers decrease in response to arsenic (Fig. 3).

4) Fig. 4A raises some doubts. First, PML bodies should be degraded upon arsenic treatment according to Fig. 3A, which is not seen here. Moreover, the p97 antibody staining

was performed after pre-extraction of the cells. Controls should be provided for specificity such as p97 depletion, or results confirmed with tagged p97. Is p97 translocation RNF4 and PML dependent?

The reviewer is quite correct in pointing out this apparent discrepancy. As we can only measure p97 recruitment when PML bodies are present we need to select fields of view where PML can still be detected. Inspection of Fig 3B reveals that PML body numbers and PML body intensity decrease by about 50% after 6 hours. It is these PML bodies that remain that are used to measure p97 recruitment. Pre-extraction is required to visualise p97 recruitment to PML bodies as p97 is highly abundant throughout the cell and unless the free soluble p97 is removed by pre-extraction it masks the material associated with PML bodies. We tested the specificity of the antibody on pre-extracted cells that had been depleted for p97 by siRNA (Supplementary Fig 2H). Similarly, we demonstrated, using siRNA to RNF4, that p97 recruitment was dependent on RNF4 (Supplementary Fig 2H). The PML dependence is tricky as we can only measure recruitment when PML is present.

5) The authors should make use of the p97 Walker B substrate trapping mutant that would support the implication of PML as substrate of p97. Can the mutant be specifically co-immunoprecipitated with PML compared to wild type p97?

We thank the reviewer for this excellent suggestion. We carried out such an experiment and indeed demonstrated that the Walker B mutation appeared to trap PML making it more evident after PML body pulldown (Fig 4C).

Minor points:

6) Labelling is done inconsistently with ATO, As, arsenic. The abbreviation ATO is not explained.

Agreed. We now use As throughout the manuscript.

7) lane 498-504: The authors elaborate on colocalisation of proteasome and p97 supporting cooperation. This seems unnecessary and inconsistent with ERAD where p97 and the proteasome segregate in different domains, shown by the Baumeister group. Also, the cited paper by Andreas Martin's group does support proximity between proteasome and p97.

We have removed the discussion of p97 and proteasome localisation.

8) The movies do not need to be published online as such.

OK. Thanks.

Reviewer #2 (Comments to the Authors (Required)):

This work aims to show that the p97/VCP segregase is required to extract SUMOylated and ubiquitinated PML from PML bodies prior to degradation by the proteasome.

Main points

1. The majority of the experiments have been carried out in a single cell model (or derivatives of). More models (including primary APLs) are needed to give mechanistic proof of the authors hypothesis.

We do not have access to primary APLs. As we carried out experiments in cell lines expressing PML alone and in patient derived cells expressing PML and PML-RARA we feel that we have established the principle that p97 is required for the arsenic induced degradation of PML and PML-RARA.

2. IP experiments should be used to corroborate proteome analyses.

This was done. See Fig 1D.

3. Proteome analyses both in U2OS as well as in APL cell based model (NB4) should be performed in presence of CB-5083.

We feel that additional proteomic experiments would not enhance the message of the paper.

4. The experiments shown in Fig 1 with exogenous proteins, should also be demonstrated with endogenous PML and also in the APL context (NB4) with PML-RARa with and without Arsenic.

We feel that additional proteomic experiments would not enhance the message of the paper.

5. The p97/VCP segregase RNA levels should be also shown in all cell models used in Fig 1.

In the validation experiments requested in point 2 above we assessed p97 protein levels (Fig 1D, inputs for p97).

6. The relevance of p97/VCP segregase should be corroborated by applying Crisp-cas or siRNA targeting and verifying the effects in presence or absence of arsenic, CB-5083.

We knocked down p97/VCP with siRNA and determined its role on PML degradation in the presence and absence of arsenic (Fig. 5A, B, C, D). We felt that knock down and pharmacological inhibition of p97 might be confusing.

7. The pharmacological inhibition with CB-5083 of p97/VCP segregase should be also shown in different cell models such as primary APL.

We did carry out pharmacological inhibition with CB-5083 in patient derived NB4 cells, but as stated above we are not a lab that handles clinical material and do not have access to primary APL cells.

8. Similarly the evaluation of the action of CB-5083 on NBs should be performed also in primary APL, APL models as well as in different models which are less artificial.

See response to point 7 above.

9. p97 cofactors UFD1 and NPLOC4 which seem to be required for arsenic induced degradation of PML are in the proteome analyses performed?

Although these cofactors are not tightly bound to p97 we did detect NPLOC4 in our proteomic analysis but did not detect UFD1.

Minor points

All raw data should be made available.

This will be done.

Reviewer #3 (Comments to the Authors (Required)):

PML and the oncogenic fusion protein PML-RAR are considered as model substrates for SUMO-dependent ubiquitylation by the SUMO-targeted ubiquitin ligase (StUbl) RNF4. Ubiquitylation and proteasomal degradation of PML and PML-RAR can be triggered by arsenic trioxide, which induces SUMO chain formation on both proteins. Here, the authors demonstrate that arsenic-induced degradation of PML and PML-RAR depends on the AAA ATPase p97/VCP and its cofactors UFD1-NPLOC4. The work was initiated by a proteomic study of purified PML nuclear bodies, in which p97 was detected as a PML associated protein, whose binding was induced upon short term treatment of cells with arsenic. This finding could be validated by immunofluorescence experiments. Further data demonstrate that the p97 inhibitor CB-5083 prevents arsenic-induced degradation of PML and PML-RAR and alters the size and morphology of PML NBs. A systematic exploration of p97 co-factors further indicate that UFD1 and NPLOC4 are involved in these processes. Altogether, the data provide convincing evidence that p97-UFD1-NPLOC4 are required for the degradation of PML and PML-RAR most likely by extracting the ubiquitylated species from PML bodies. These findings are in line with previous and very recent reports describing an involvement of p97 in extracting mammalian SUMOylated/ubiquitylated StUBL targets, such as FANCI or PARP1, from chromatin (Gibbs-Seymour et al., Mol Cell, 2015; Krastev et al., Nature Cell Biology 2022). Similar observations have also been made on the p97 orthologue CDC48 in lower eukaryotes, such as fission and baker's yeast. While this takes away some originality, the finding that p97 is involved in extracting proteins also from PML bodies and is required for PML degradation is novel and of considerable significance. Overall, the data are mostly convincing and support the conclusions. Nevertheless, I do have a few comments and suggestions that might further strengthen the work.

Major points:

1) As mentioned above, the work was initiated by a proteomic study of purified PML nuclear bodies under control conditions or upon treatment of cells with arsenic trioxide. Purification of intact PML NBs has been a major challenge due to the insolubility of these structure. Only very recently, proximity-based proteomics have facilitated the identification of PML body components in a system-wide manner (Barroso-Gomila, Nature communications, 2021). In the current work the authors performed a biochemical purification of PML bodies, which is based on subcellular fractionation and final affinity purification on anti GFP nanobody beads using YFP-PML as a bait. Using this approach, the authors identified around 200 PML associated proteins under basal conditions and defined a set of 21 proteins, whose association with PML bodies is enhanced upon arsenic treatment (Figure 1C). This set contains p97/VCP as well as a number of bona fide PML body components, such as Topors, PIAS1/3, SUMO etc. Most of the proteins are annotated as NB-resident proteins validating the approach. Furthermore, there is also a fairly good overlap with the data obtained by proximity-based proteomics.

These are good points. We have compared our data with that obtained by proximity labelling (Barroso-Gomila et al., 2021) and to proteins colocalising with PML bodies in the Human Protein Atlas (Thul et al., 2017). The full comparisons are now included in the Supplementary data file 1 and the overlap with our arsenic induced list is now indicated in Fig 1C. In addition we carried out Western blot analysis of PML body associated proteins (Fig. 1D) to confirm a number of our mass spec identifications.

Interestingly, however, the authors also identified a set of nucleolar proteins, such as MDN1, PELP1 or SIMC1, which were so far not reported to reside in PML bodies. It would be important to validate that these proteins are indeed associated with PML bodies under basal conditions and/or upon treatment with arsenic. To my opinion, this is important to exclude that the purification is partially contaminated with nucleoli. Alternatively, SIMC1 or MDN1 may associate with SUMOylated PML only after cell lysis due to the presence of characterized SUMO binding motifs in both proteins.

Even though the focus of the work is on p97 rather than any of the other identified proteins, this is not relevant for any of the follow-up experiments on p97. Nevertheless, this point should be addressed or a post-lysis artefact should at least be discussed.

We agree that this is an interesting point. In fact the Barroso-Gomila et al., 2021 proximity labelling paper also identified SIMC1 as a PML body component. We compared our PML list with nucleolar components identified in the Human Protein Atlas and there is a significant overlap. The full comparisons are now included in the Supplementary data file 1. We do not believe that this is a result of nucleolar contamination or post lysis association of nucleoli. If this was the case, we would expect to see major nucleolar proteins associated with our PML preps and this is not what we see. Rather, we believe this reflects a dynamic association of some nucleolar proteins with PML bodies. This is suggested by the proximity labelling based identification of SIMC1 as a PML associated protein. Moreover, the immunofluorescence experiment with PELP1 and MDN1 suggested by the reviewer that shows SUMO target PELP1 and SUMO Interaction Motif (SIM) containing MDN1 are present in nucleoli with a fraction also colocalising with PML bodies (Supplementary Fig. 2G). There is also a body of literature supporting the idea of proteins shuttling between PML bodies and nucleoli (Imrichova et al., Dynamic PML protein nucleolar associations with persistent DNA damage lesions in response to nucleolar stress and senescence-inducing stimuli. *Aging* . 2019; 11:7206-7235; Shim et al., The autophagic protein LC3 translocates to the nucleus and localizes in the nucleolus associated to NUFIP1 in response to cyclic mechanical stress. *Autophagy*. 2020; 16:1248-1261; Souquere et al., Comparative ultrastructure of CRM1-Nucleolar bodies (CNoBs), Intranucleolar bodies (INBs) and hybrid PML/p62 bodies uncovers new facets of nuclear body dynamic and diversity. *Nucleus*. 2015; 6:326-38; Sharma et al., N4BP1 is a newly identified nucleolar protein that undergoes SUMO-regulated polyubiquitylation and proteasomal turnover at promyelocytic leukemia nuclear bodies. *J Cell Sci*. 2010; 123:1227-34). It seems likely that certain nucleolar proteins that are either SUMO modified or contain SIMs can partition between nucleoli and PML bodies.

2) While ubiquitin binding of the mammalian p97-UFD1-NPLOC4 complex is quite well understood, it is still not entirely clear whether SUMOylation of substrates or hybrid SUMO-UB chains can promote binding. To see whether ubiquitylation is strictly required for recruitment of p97 to PML bodies, cells could be co-treated with arsenic and an Ub E1 inhibitor, such as TAK-243.

We carried out this experiment. This shows that TAK243 does indeed block recruitment of p97 into PML bodies indicating that ubiquitylation is strictly required for arsenic induced recruitment of p97 into PML bodies (Supplementary Fig. 2H).

3) Gibbs-Seymour et al. (*Mol Cell*, 2015) reported that p97 cooperates with the cofactor DVC1/SPRTN in the extraction of SUMOylated/Ubiquitylated FANCI/FANCD2 from chromatin. Based on these findings it would be interesting to see whether and how depletion of DVC1/SPRTN affect PML degradation and PML morphology. This could be particularly interesting in case K63 chains are detected on PML (see below).

We carried out the experiment to determine if K63 chains were associated with PML bodies and this was indeed the case (Fig 6F). We also used siRNA to deplete DVC1/SPRTN, but this did not block recruitment of p97 into PML bodies (Supplementary Fig. 2H).

4) As shown previously, endogenous PML (or at least some isoforms) are relatively resistant to arsenic induced degradation (also when compared to PML-RAR) under conditions of endogenous RNF4 expression (Figure 2D). Nevertheless, the pattern of higher molecular weight-bands is indicative of SUMOylation-ubiquitylation. This raises the question whether some ubiquitylated species on PML are non-proteolytic. To address this issue, immunoblotting in Figure 6D should be done with Ub K63/K48 linkage-specific antibodies in addition to the pan-Ub antibody FK2.

We thank the reviewer for this suggestion. We carried out this experiment and the result was very interesting. Blotting with the K48 specific antibody indicated that K48 chains associated with PML were only detected in the presence of the p97 inhibitor (Fig. 6E) suggesting that they are a short-lived intermediate that is very rapidly transited via p97 to the proteasome. Although undetected in the Western blot experiment we did observe an arsenic induced increase in K48 chains in the absence of p97 inhibition by mass spec (Fig. 6G). We also detected association of K63 chains with PML, but this was not stabilised by the p97 inhibitor (Fig. 6F). K63 chains associated with PML were detected by mass spec but the arsenic induced increase was not statistically significant (Fig. 6G).

5) Gibbs-Seymour et al., Mol Cell, 2015; Krastev et al., Nature Cell Biology 2022 should be cited.

These papers have been cited in the Discussion.

Gibbs-Seymour I, Oka Y, Rajendra E, Weinert BT, Passmore LA, Patel KJ, Olsen JV, Choudhary C, Bekker-Jensen S, Mailand N. Ubiquitin-SUMO circuitry controls activated fanconi anemia ID complex dosage in response to DNA damage. Mol Cell. 2015, 57:150-64.

Krastev DB, Li S, Sun Y, Wicks AJ, Hoslett G, Weekes D, Badder LM, Knight EG, Marlow R, Pardo MC, Yu L, Talele TT, Bartek J, Choudhary JS, Pommier Y, Pettitt SJ, Tutt ANJ, Ramadan K, Lord CJ. The ubiquitin-dependent ATPase p97 removes cytotoxic trapped PARP1 from chromatin. Nat Cell Biol. 2022, 24:62-73.

December 21, 2022

RE: JCB Manuscript #202201027R

Dr. Ronald T Hay
University of Dundee
Centre for Gene Regulation and Expression Sir James Black Centre Dow Street
Dundee, Scotland DD15EH
United Kingdom

Dear Dr. Hay:

Thank you for submitting your revised manuscript entitled "The p97/VCP segregase is required for arsenic induced degradation of PML and the PML-RARA oncoprotein". We would be happy to publish your paper in JCB pending final revisions necessary to meet our formatting guidelines (see details below). In your final revision, please be sure to address the lingering concerns of reviewer #1 with appropriate text edits. In light of the technical difficulties expressed by the authors in obtaining the requested colocalization/co-IP experiments by reviewer #2, and after consulting with this reviewer, these data would not be required for provisional acceptance.

To avoid unnecessary delays in the acceptance and publication of your paper, please read the following information carefully. Please go through all the formatting points paying special attention to those marked with asterisks.

A. MANUSCRIPT ORGANIZATION AND FORMATTING:

Full guidelines are available on our Instructions for Authors page, <https://jcb.rupress.org/submission-guidelines#revised>.
Submission of a paper that does not conform to JCB guidelines will delay the acceptance of your manuscript.

1) Text limits: Character count for Articles and Tools is < 40,000, not including spaces. Count includes title page, abstract, introduction, results, discussion, and acknowledgments. Count does not include materials and methods, figure legends, references, tables, or supplemental legends.

2) Figures limits: Articles and Tools may have up to 10 main text figures.

*** Please note that main text figures should be provided as individual, editable files.

3) Figure formatting:

Molecular weight or nucleic acid size markers must be included on all gel electrophoresis.

Scale bars must be present on all microscopy images, including inset magnifications.

Also, please avoid pairing red and green for images and graphs to ensure legibility for color-blind readers. If red and green are paired for images, please ensure that the particular red and green hues used in micrographs are distinctive with any of the colorblind types. If not, please modify colors accordingly or provide separate images of the individual channels.

4) Statistical analysis:

Error bars on graphic representations of numerical data must be clearly described in the figure legend.

*** The number of independent data points (n) represented in a graph must be indicated in the legend -please add 'n' for Fig 4B. Please, indicate whether 'n' refers to technical or biological replicates (i.e. number of analyzed cells, samples or animals, number of independent experiments) -please, revise Fig 6G.

If independent experiments with multiple biological replicates have been performed, we recommend using distribution-reproducibility SuperPlots (please, see Lord et al., JCB 2020) to better display the distribution of the entire dataset, and report statistics (such as means, error bars, and P values) that address the reproducibility of the findings.

*** Statistical methods should be explained in full in the materials and methods in a separate section.

For figures presenting pooled data the statistical measure should be defined in the figure legends.

Please also be sure to indicate the statistical tests used in each of your experiments (both in the figure legend itself and in a separate methods section) as well as the parameters of the test (for example, if you ran a t-test, please indicate if it was one- or two-sided, etc.).

*** As you used parametric tests in your study (i.e. t-tests), you should have first determined whether the data was normally distributed before selecting that test. In the stats section of the methods, please indicate how you tested for normality. If you did not test for normality, you must state something to the effect that "Data distribution was assumed to be normal but this was not formally tested."

5) Abstract and title:

The abstract should be no longer than 160 words and should communicate the significance of the paper for a general audience.

The title should be less than 100 characters including spaces. Make the title concise but accessible to a general readership.

6) Materials and methods:

Should be comprehensive and not simply reference a previous publication for details on how an experiment was performed. The text should not refer to methods "...as previously described."

Also, the materials and methods should be included with the main manuscript text and not in the supplementary materials.

7) For all cell lines, vectors, constructs/cDNAs, etc. - all genetic material: please include database / vendor ID (e.g., Addgene, ATCC, etc.) or if unavailable, please briefly describe their basic genetic features, even if described in other published work or gifted to you by other investigators (and provide references where appropriate).

*** Please be sure to provide the sequences for all of your oligos: primers, si/shRNA, RNAi, gRNAs, etc. in the materials and methods.

You must also indicate in the methods the source, species, and catalog numbers/vendor identifiers (where appropriate) for all of your antibodies, including secondary. If antibodies are not commercial please add a reference citation if possible.

8) Microscope image acquisition:

The following information must be provided about the acquisition and processing of images:

- a. Make and model of microscope
- b. Type, magnification, and numerical aperture of the objective lenses
- c. Temperature
- d. imaging medium
- e. Fluorochromes
- f. Camera make and model
- g. Acquisition software
- h. Any software used for image processing subsequent to data acquisition. Please include details and types of operations involved (e.g., type of deconvolution, 3D reconstitutions, surface or volume rendering, gamma adjustments, etc.).

10) Supplemental materials:

There are strict limits on the allowable amount of supplemental data. Articles/Tools may have up to 5 supplemental figures. There is no limit for supplemental tables.

*** Please note that supplemental figures and tables should be provided as individual, editable files.

*** A summary of all supplemental material should appear at the end of the Materials and Methods section (please see any recent JCB paper for an example of this summary).

11) Video legends: Should describe what is being shown, the cell type or tissue being viewed (including relevant cell treatments, concentration and duration, or transfection), the imaging method (e.g., time-lapse epifluorescence microscopy), what each color represents, how often frames were collected, the frames/second display rate, and the number of any figure that has related video stills or images.

12) eTOC summary:

A ~40-50 word summary that describes the context and significance of the findings for a general readership should be included on the title page.

*** The statement should be written in the present tense and refer to the work in the third person. It should begin with "First author name(s) et al..." to match our preferred style.

13) Conflict of interest statement:

*** JCB requires inclusion of a statement in the acknowledgements regarding competing financial interests. If no competing financial interests exist, please include the following statement: "The authors declare no competing financial interests."

14) A separate author contribution section is required following the Acknowledgments in all research manuscripts.

*** All authors should be mentioned and designated by their first and middle initials and full surnames and the CRediT nomenclature is encouraged (<https://casrai.org/credit/>).

15) ORCID IDs: ORCID IDs are unique identifiers allowing researchers to create a record of their various scholarly contributions in a single place. At resubmission of your final files, please consider providing an ORCID ID for as many contributing authors as possible.

16) Materials and data sharing:

All animal and human studies must be conducted in compliance with relevant local guidelines, such as the US Department of Health and Human Services Guide for the Care and Use of Laboratory Animals or MRC guidelines, and must be approved by the authors' Institutional Review Board(s). A statement to this effect with the name of the approving IRB(s) must be included in the Materials and Methods section.

*** As a condition of publication, authors must make protocols and unique materials (including, but not limited to, cloned DNAs; antibodies; bacterial, animal, or plant cells; and viruses) described in our published articles freely available upon request by researchers, who may use them in their own laboratory only. All materials must be made available on request and without undue delay. We strongly encourage to deposit all the cell lines/strains and reagents generated in this study in public repositories.

All datasets included in the manuscript must be available from the date of online publication, and the source code for all custom computational methods, apart from commercial software programs, must be made available either in a publicly available database or as supplemental materials hosted on the journal website. Numerous resources exist for data storage and sharing (see Data Deposition: <https://rupress.org/jcb/pages/data-deposition>), and you should choose the most appropriate venue based on your data type and/or community standard. If no appropriate specific database exists, please deposit your data to an appropriate publicly available database.

17) Please note that JCB now requires authors to submit Source Data used to generate figures containing gels and Western blots with all revised manuscripts. This Source Data consists of fully uncropped and unprocessed images for each gel/blot displayed in the main and supplemental figures. The Source Data files will be directly linked to specific figures in the published article.

Since your paper includes cropped gel and/or blot images, please be sure to provide one Source Data file for each figure that contains gels and/or blots along with your revised manuscript files. File names for Source Data figures should be alphanumeric without any spaces or special characters (i.e., SourceDataF#, where F# refers to the associated main figure number or SourceDataFS# for those associated with Supplementary figures). The lanes of the gels/blots should be labeled as they are in the associated figure, the place where cropping was applied should be marked (with a box), and molecular weight/size standards should be labeled wherever possible.

B. FINAL FILES:

Thank you for this interesting contribution, we look forward to publishing your paper in Journal of Cell Biology.

Sincerely,

Pier Paolo Di Fiore, MD, PhD
Editor
The Journal of Cell Biology

Lucia Morgado-Palacin, PhD
Scientific Editor
Journal of Cell Biology

Reviewer #1 (Comments to the Authors (Required)):

The authors fully addressed my points. This is a carefully conducted and very relevant study clearly worth publishing. Given the mechanistic outline, I would agree with the authors that further evaluation in patient cells would go beyond the scope.

minor editing suggestions:

The expression "is required" in the title does not seem to do justice to the active role of p97 uncovered in this study.

Lane 286: state in the text that requirement in S2H was shown by siRNA.

Lane 310: state that codepletion of UFD1 is "as expected".

Fig 4C: mark "p97-myc" so that the anti-myc western is self-explanatory

Fig. s5: label NT and legend do not match.

Reviewer #2 (Comments to the Authors (Required)):

In the present manuscript, the authors aim to investigate on the processes of PML and PML-RAR modification. They have performed proteomics on PML bodies revealing increased association of p97/VCP segregase with PML bodies after arsenic treatment. Pharmacological inhibition of p97 altered the number, morphology and size of PML bodies, accumulated SUMO and ubiquitin modified PML and blocked arsenic induced degradation of PML-RARA and PML. p97 localized to PML bodies in response to arsenic and siRNA mediated depletion showed that p97 cofactors UFD1 and NPLOC4 were critical for PML degradation.

The manuscript is clearly written.

The authors have replied to nearly all R1/2/3 suggestions.

I still think that some experiments in primary APLs would have been interesting, but I see that the authors do not have access to those materials despite, I must say, they certainly might have been collaborating with other groups having this cells. It also would have been possible to use cells derived from APL models.

This being said, I would suggest at least to see that p97 co-localises (and/or it is IPed) with PML-RAR upon arsenic treatment in NB4-APL cells.

To me this is important to corroborate all data in a system, which is more near to the 'real' APL pathology and less artificial.

Reviewer #3 (Comments to the Authors (Required)):

In their revised version the authors have addressed all major concerns that I had raised on the initial version of the manuscript. The new experimental data provided by the authors further strengthen the work. To my opinion, the manuscript is now acceptable for publication in JCB.